# Pt nanoshells with a high NIR-II photothermal conversion efficiency mediates multimodal neuromodulation against ventricular arrhythmias

Chenlu Wang[1,10], Liping Zhou[2,3,4,5,6,7,8,10], Chengzhe Liu[2,3,4,5,6,7,8,10], Jiaming Qiao[2,3,4,5,6,7,8], Xinrui Han[2,3,4,5,6,7,8], Luyang Wang[1], Yaxi Liu[1], Bi Xu[1], Qinfang Qiu[2,3,4,5,6,7,8], Zizhuo Zhang[2,3,4,5,6,7,8], Jiale Wang[2,3,4,5,6,7,8], Xiaoya Zhou[2,3,4,5,6,7,8] ✉, Mengqi Zeng [1], Lilei Yu[2,3,4,5,6,7,8] ✉ & Lei Fu [1,7,8,9] ✉

Autonomic nervous system disorders play a pivotal role in the pathophysiology of cardiovascular diseases. Regulating it is essential for preventing and treating acute ventricular arrhythmias (VAs). Photothermal neuromodulation is a nonimplanted technique, but the response temperature ranges of transient receptor potential vanilloid 1 (TRPV1) and TWIK-related K+ Channel 1 (TREK1) exhibit differences while being closely aligned, and the acute nature of VAs require that it must be rapid and precise. However, the low photothermal conversion efficiency (PCE) still poses limitations in achieving rapid and precise treatment. Here, we achieve a nearly perfect blackbody absorption and a high PCE in the second near infrared (NIR-II) window (73.7% at 1064 nm) via a Pt nanoparticle shell (PtNP-shell). By precisely manipulating the photothermal effect, we successfully achieve rapid and precise multimodal neuromodulation encompassing neural activation (41.0–42.9 °C) and inhibition (45.0–46.9 °C) in a male canine model. The NIR-II photothermal modulation additionally achieves multimodal reversible autonomic modulation and confers protection against acute VAs associated with myocardial ischemia and reperfusion injury in interventional therapy.

Cardiovascular disease has emerged as a leading cause of mortality, with acute myocardial infarction being one of the most pernicious ailments[1,2]. Myocardial ischemia (MI) frequently precipitates acute ventricular arrhythmias (VAs), impeding prompt and efficacious treatment for acute myocardial infarction. Furthermore, conventional interventional procedures for MI are unable to circumvent concomitant myocardial reperfusion injury and associated VAs[3]. The autonomic nervous system, encompassing sympathetic and parasympathetic nerves, plays a role in cardiovascular modulation; both are naturally antagonistic. Sympathetic inhibition or parasympathetic

[1]College of Chemistry and Molecular Sciences, Wuhan University, Wuhan, China. [2]Department of Cardiology, Renmin Hospital of Wuhan University, Wuhan, China. [3]Hubei Key Laboratory of Autonomic Nervous System Modulation, Wuhan, China. [4]Cardiac Autonomic Nervous System Research Center of Wuhan University, Wuhan, China. [5]Hubei Key Laboratory of Cardiology, Wuhan, China. [6]Cardiovascular Research Institute, Wuhan University, Wuhan, China. [7]Taikang Center for Life and Medical Sciences, Wuhan University, Wuhan, China. [8]Institute of Molecular Medicine, Renmin Hospital of Wuhan University, Wuhan, China. [9]The Institute for Advanced Studies, Wuhan University, Wuhan, China. [10]These authors contributed equally: Chenlu Wang, Liping Zhou, Chengzhe Liu. ✉e-mail: whuzhouxiaoya@whu.edu.cn; lileiyu@whu.edu.cn; leifu@whu.edu.cn

activation has been shown to stabilize cardiac electrophysiology, safeguard against MI, and reduce the incidence of VAs[4]. However, cardiac sympathetic denervation (CSD), stellate ganglion block (SGB), and renal denervation (RDN) are associated with certain adverse effects, including Horner's syndrome[5], inadvertent bleeding, and inconsistent ablation outcomes[6]. Both conventional vagus nerve stimulation (VNS) and optogenetic neuromodulation necessitate the implantation of in vivo electrical stimulation[7] or light source[8] devices. Furthermore, optogenetic neuromodulation requires viral transfection of photosensitive proteins[8], thereby limiting the clinical advancement of these therapeutic approaches.

In recent years, several studies have demonstrated that light-activated nanotransducers can induce local heating effects, leading to the activation or inhibition of nerves[9–11]. This discovery is attributed to the identification of temperature-sensitive ion channels in neurons, such as transient receptor potential vanilloid 1 (TRPV1)[12] and TWIK-related K⁺ Channel 1 (TREK1)[13]. The activation of specific temperature-sensitive ion channels necessitates precise temperature ranges[12–14]. Considering the acute nature of neural responses, a therapeutic strategy with rapid and accurate modulation is required. The light in the second near-infrared window (NIR-II, 900–1700 nm) has reduced tissue scattering and absorption and increased maximum permissible exposure (MPE) for biological tissues compared to the light in the first near-infrared (NIR-I, 650–900 nm) and visible window[15]. Consequently, this enables non-invasive and non-implantable neuromodulation using the NIR-II photothermal effect. However, its neural response rate and accuracy are currently limited by low photothermal conversion efficiency (PCE).

In this work, we report a near blackbody NIR-II Pt nanoparticle shell (PtNP-shell) for protection against MI and myocardial reperfusion injury accompanying intervention. The PtNP-shell, synthesized through a simple electrocoupling substitution reaction using liquid metal nanoparticles as templates (Fig. 1a), possesses surface pores and a hollow structure. It demonstrates a nearly perfect blackbody absorption, enhanced absorption of light, and then a high PCE in the NIR-II window (73.7% at 1064 nm). By leveraging the local heating effect mediated by PtNP-shell, we achieve rapid, efficient, and precise multimodal autonomic neuromodulation. Specifically, parasympathetic activation and sympathetic inhibition are accomplished by activating TRPV1 (41.0–42.9 °C) and TREK1 (45.0–46.9 °C) channels, respectively. Photothermal autonomic neuromodulation mediated by PtNP-shell effectively stabilizes cardiac electrophysiology and reduces VAs incidence in both myocardial ischemia-reperfusion (I/R) injury model and MI model, respectively (Fig. 1b).

## Results

### Synthesis and characterization of PtNP-shell

The PtNP-shell was synthesized through an electrocoupling substitution reaction between chloroplatinate and Ga nanoparticles (GaNPs). Ga nanoparticles were obtained by sonication of pure metal Ga. To achieve a balanced particle size and oxidation degree of GaNPs, pure gallium was sequentially sonicated in ethanol and water for 30 min to obtain gallium nanoparticles with reduced oxidation (Supplementary Fig. 1a). In accordance with the electrochemical redox potential of the redox couple (Ga³⁺/Ga: −0.529 V; PtCl₆²⁻/PtCl₄²⁻: 0.726 V; PtCl₄²⁻/Pt:

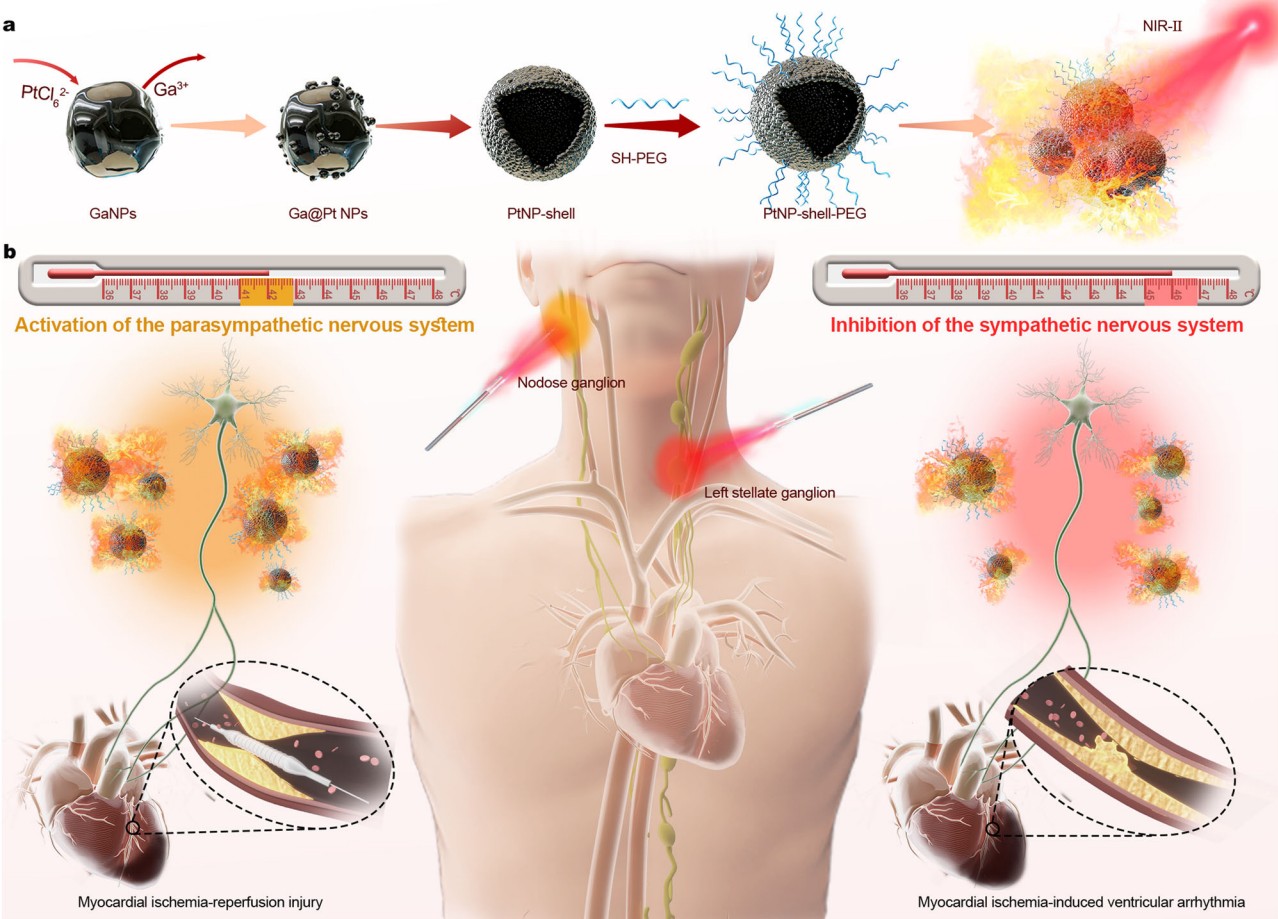

**Fig. 1 | The synthesis steps of the PtNP-shell and the concept of mediating precise photothermal effects against ventricular arrhythmias. a** The synthesis steps of PtNP-shell and schematic diagram of photothermal effect. **b** Schematic diagram of multimodal autonomic modulation mediated by photothermal effect of PtNP-shell for the treatment of myocardial ischemia-reperfusion injury and myocardial ischemia-induced ventricular arrhythmias.

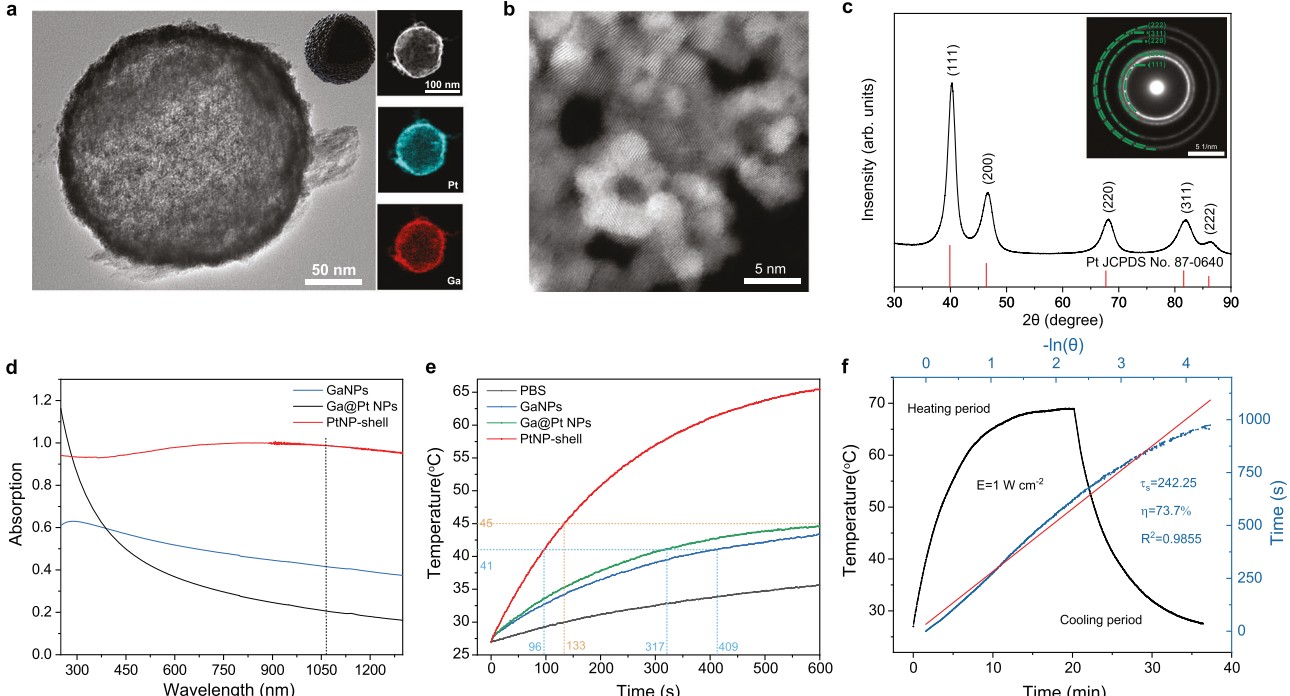

**Fig. 2 | Characterization of PtNP-shell. a** TEM image of PtNP-shell, $n = 10$ independent replicates (inset: schematic diagram of PtNP-shell; right: element mapping). **b** STEM images of PtNP-shell surface, $n = 6$ independent replicates. **c** XRD spectrum of PtNP-shell (inset: SAED pattern). **d** UV–vis–NIR absorption spectrum of GaNPs, Ga@Pt NPs, and PtNP-shell (75 µg mL⁻¹). **e** Temperature elevation curves of PBS, GaNPs, Ga@Pt NPs and PtNP-shell (50 µg mL⁻¹) under NIR-II laser irradiation (1 W cm⁻²). **f** Calculation of the PCE at 1064 nm (PtNP-shell: 50 µg mL⁻¹). Source data are provided as a Source Data file.

0.758 V)[16,17], Pt (IV) can be in situ reduced by Ga and deposited on the surface of GaNPs to form a core-shell structure (Supplementary Fig. 1b, c). The hollow PtNP-shell is synthesized after completion of the reaction (Fig. 2a). Simultaneously with the reduction of Pt (IV), Ga oxide is formed, creating the skeleton of the PtNP-shell (right in Fig. 2a). The surface of the PtNP-shell exhibits a rough texture (Supplementary Fig. 2). The scanning transmission electron microscopy (STEM) images reveal numerous irregular and uneven pores on its surface (Supplementary Fig. 3a) and PtNP-shell is composed of Pt nanoparticles (PtNPs) with 2–5 nm (Fig. 2b). High-resolution TEM (HRTEM) image is acquired to character the structure of PtNPs. As shown in Supplementary Fig. 3b, PtNPs exhibit a single crystal structure with a lattice stripe spacing of 0.23 nm corresponding to the (111) crystal plane. Meanwhile, the corresponding Fast Fourier Transform (FFT) pattern (inset in Supplementary Fig. 3b) shows the typical diffraction patterns of face-centered cubic structure along [111] zone axis.

In the X-ray powder diffraction (XRD) spectrogram result (Fig. 2c), all peaks can be attributed to the crystal phase of Pt (JCPDS: 87-0640), consistent with the selected area electron diffraction (SAED) pattern findings (inset in Fig. 2c). However, no peaks corresponding to gallium oxide were observed in the XRD spectrogram, possibly due to its low content. The XRD spectrogram (Supplementary Fig. 4) of PtNP-shell prior to reacting with KOH showed that the gallium oxide contained in PtNP-shell was GaOOH (JCPDS: 06-0180). X-ray photoelectron spectroscopy (XPS) analysis reveals the presence of Ga and O in GaNPs, while Ga@Pt NPs and PtNP-shell exhibit the coexistence of Ga, O, and Pt (Supplementary Fig. 5). As depicted in the Supplementary Fig. 6, despite treatment with KOH, no presence of K element was detected in the PtNP-shell. The strong signals of Pt $4f_{5/2}$ and Pt $4f_{7/2}$ indicate that the Pt in Ga@Pt NPs and PtNP-shell exists in a metallic state[18]. In GaNPs, the peak centered at 1118.11 eV is attributed to Ga³⁺ $2p_{3/2}$, while the peak centered at 1115.89 eV corresponds to Ga $2p_{3/2}$. In Ga@Pt NPs, the peak centered at 1118.80 eV is assigned to Ga³⁺ $2p_{3/2}$, and the peak centered at 1116.52 eV corresponds to Ga $2p_{3/2}$. As for PtNP-shell, the presence of a peak around 1118.56 eV indicates complete oxidation of Ga in PtNP-shell into Ga³⁺[16]. The O $1s$ spectrum was fitted using two peak functions, which were assigned to Ga–O at 530.44 eV (GaNPs), 530.98 eV (Ga@Pt NPs), 531.74 eV (PtNP-shell) and Ga–OH at 531.71 eV (GaNPs), 532.08 eV (Ga@Pt NPs), 532.74 eV (PtNP-shell)[19]. Compared to GaNPs and Ga@Pt NPs, the binding energy of the Ga–O and Ga–OH peaks in the PtNP-shell is shifted toward higher values, indicating a lower oxidation degree of the PtNP-shell. PtNP-shell was treated with KOH (0.67 M) to reduce the gallium oxide content and the surface potential was reduced from 45.8 to −25.7 mV, and then encapsulated with methoxypoly(ethylene glycol) thiol (mPEG-SH₅₀₀₀) to enhance its biocompatibility and the surface potential was changed to −19.9 mV (Supplementary Table. 1). The statistically averaged hydrated nanoparticle size of PtNP-shell based on the dynamic light scattering diagram was 200.1 nm with a uniform size distribution, indicating the nanoparticle was well dispersed in water (Supplementary Fig. 7). After standing for different days, the statistically averaged hydrodynamic size of PtNP-shell was determined using dynamic light scattering (Supplementary Fig. 8). It is worth noting that the change of the average hydrated nanoparticle size of PtNP-shell remains negligible after 14 days, indicating its excellent stability.

**Blackbody absorption and photothermal property of PtNP-shell**
Due to the presence of pores and a hollow structure in the PtNP-shell, light propagating in the space bounces at the rough surface of PtNP-shell until it encounters one of the pores, where it continues to bounce within the PtNP-shell. The random distribution of these pores results in completely random light reflection, akin to Brownian motion[20]. Consequently, the probability of light escaping from other pores is extremely low, rendering PtNP-shell behave like a blackbody and produce an efficient infrared heater[21–23]. This enhanced absorption of light by PtNP-shell exhibits nearly perfect blackbody absorption characteristics (Supplementary Fig. 9a). In the range of 250–1300 nm, the PtNP-shell exhibits an absorbance close to 1 at 75 µg mL⁻¹, which is significantly

enhanced in the range of 400–1300 nm compared to GaNPs and Pt-coated Ga nanoparticles (Ga@Pt NPs) (Fig. 2d). According to the Lambert–Beer law (A/L = εC, where ε is the extinction coefficient), a linear relationship between absorption intensity (at 1064 nm) and concentration was established, with an extinction coefficient measured as $13.3 \, L g^{-1} cm^{-1}$ at 1064 nm (Supplementary Fig. 9b). Varying concentrations of PtNP-shell resulted in different shades of gray being generated, with significantly darker grayness observed under identical conditions compared to GaNPs and Ga@Pt NPs (Supplementary Fig. 10a). These distinctive features were characterized by their respective positions within an RGB cube representation, wherein on the diagonal connecting darkest and brightest points, PtNP-shell was found closer to the darkest point than both other materials (Supplementary Fig. 10b).

The photothermal properties of PtNP-shell were verified by irradiating the dispersion of PtNP-shell in water with NIR-II light at 1064 nm ($1 \, W cm^{-2}$). Even in vitro, PtNP-shell ($50 \, \mu g \, mL^{-1}$) exhibited rapid temperature elevation, achieving a rise from room temperature to 41.0 and 45.0 °C within only 96 and 133 s, respectively. However, for GaNPs (409 s and over 600 s) and Ga@Pt NPs (317 s and over 600 s), it took significantly longer time to reach the same temperatures (Fig. 2e). The corresponding thermal images of the PtNP-shell with different concentrations under different irradiation times are shown in Supplementary Fig. 11. The heating effect of the PtNP-shell ($50 \, \mu g \, mL^{-1}$) gradually increased the ΔT from 7.72 to 52.17 °C when exposed to NIR-II laser for a duration of 600 s while varying the optical power density at 1064 nm between 0.25 and $1.5 \, W cm^{-2}$ (Supplementary Fig. 12). The heating rate of the SH-PEG modified PtNP-shell is significantly higher compared to that of the unmodified PtNP-shell (Supplementary Fig. 13a), potentially attributed to the agglomeration tendency of unmodified PtNP-shell at elevated temperatures, leading to a reduction in photothermal performance. Following 600 s of laser irradiation at 1064 nm, the statistically averaged hydrodynamic size for SH-PEG-modified PtNP-shell was measured as 206.5 nm (Supplementary Fig. 13b), whereas unmodified PtNP-shell exhibited a size of 1517 nm (Supplementary Fig. 13c). TEM analysis further confirmed the observed agglomeration behavior in unmodified PtNP-shell subsequent to laser irradiation (Supplementary Fig. 13). The PCE of PtNP-shell was quantified as 73.7% when balancing the energy input from photons with heat dissipation within the system (Fig. 2f), representing a high PCE at 1064 nm (Supplementary Table 2). These results indicate that PtNP-shell exhibits excellent photothermal performance in the NIR-II window. Additionally, no significant changes in temperature or morphology were observed even after five cycles of irradiation (Supplementary Fig. 14), suggesting exceptional photothermal stability.

### The photothermal effect of PtNP-shell enables precise modulation of neurons in vitro

To investigate the photothermal effects of PtNP-shell on neuronal activity at multiple levels, we conducted calcium imaging experiments in hippocampal neuron (HT-22) cells (Fig. 3a, b). The immunoblotting results revealed abundant expression of both TRPV1 and TREK1 ion channels in HT-22 cells (Fig. 3c). The direct effect of PtNP-shell on the excitability of these two different ion channels was assessed under NIR-II irradiation using a calcium ion indicator (Fluo-4 AM). Upon NIR-II laser irradiation, the temperature of the PtNP-shell (+) group increased compared to that of the PtNP-shell (−) group, resulting in a significantly higher percentage of responding cells (Fig. 3d) ($p < 0.001$). The micrographs fluorescence intensity curve of HT-22 neurons cultured with PtNP-shell showed significant $Ca^{2+}$ influx upon NIR-II laser irradiation for 35 ± 5 s and after the temperature reached 42.0 °C (Fig. 3e). In contrast, application of NIR-II laser irradiation with phosphate-buffered saline (PBS) did not induce significant $Ca^{2+}$ influx.

Subsequently, neuronal excitation was induced and calcium signals were increased by perfusion of 15 mM KCl in the PtNP-shell (−) group and PtNP-shell (+) group ($50 \, \mu g \, mL^{-1}$), respectively. This phenomenon can be attributed to the elevation of extracellular potassium ion concentration, which triggers neuronal depolarization and subsequently leads to a substantial increase in intracellular calcium ion concentration[24]. Under NIR-II laser irradiation, the proportion of HT-22 cells responding to high-concentration KCl stimulation was significantly lower in the PtNP-shell (+) group compared to that in the PtNP-shell (−) group at ~46.0 °C (Fig. 3f). The difference may be due to the activation of the TERK1 ion channel in the PtNP-shell (+) group, which can induce neuronal hyperpolarization and make intracellular and extracellular calcium ion concentrations tend to recover[25]. Interestingly, the PtNP-shell influenced the fluorescence intensity of HT-22 cells not with a sustained decrease but with an initial rise followed by a subsequent decrease (Fig. 3g). This observation may be associated with the activation of TRPV1 channel at around 42.0 °C[9]. With increasing temperature, TRPV1 and TREK1 channels were sequentially activated. These findings suggest that PtNP-shell can achieve precise temperature control within a short duration through its own high PCE for both neuronal excitation and inhibition.

Cytotoxicity assays were then conducted to investigate the potential neurotoxicity of PtNP-shell application. As shown in Fig. 3h, concentrations of PtNP-shell below $100 \, \mu g \, mL^{-1}$ exhibited no significant toxic effects on HT-22 cells. Even when the concentration of PtNP-shell was increased to $200 \, \mu g \, mL^{-1}$, the survival rate of neuronal cells remained approximately at 52.11%. After neurons were co-cultured with PtNP-shell ($50 \, \mu g \, mL^{-1}$) for 24 h, cross-sectional TEM and SEM images showed that PtNP-shell was randomly distributed inside or on the surface of the cells (Supplementary Fig. 15). This is attributed to the fact that the PtNP-shell exhibits a particle size of ~200 nm, enabling smaller particles to traverse the cell membrane. Furthermore, the impact of PtNP-shell photothermal stimulation parameters on cell viability was assessed through analysis of HT-22 cell survival under NIR-II laser irradiation. Notably, when a concentration of $50 \, \mu g \, mL^{-1}$ PtNP-shell and an NIR-II laser with a power density of $0.75 \, W cm^{-2}$ was applied for a brief duration, the survival rate exceeded 91.87% for HT-22 cells (Fig. 3i). Even with an increase in power density to $1 \, W cm^{-2}$, the survival rate for HT-22 cells remained around 82.71% after 90 s of irradiation (Fig. 3j). These results indicate that PtNP-shell does not induce significant damage to neurons under controlled NIR-II laser irradiation.

### PtNP-shell photothermal activation of the parasympathetic nervous system

Western blotting analysis of peripheral ganglia from the canine autonomic nervous system revealed the expression of TRPV1 and TREK1 heat-sensitive ion channels in both the nodose ganglion (NG) and the left stellate ganglion (LSG). Notably, TRPV1 was abundantly expressed in the NG of the parasympathetic nervous system, while TREK1 exhibited higher levels in the LSG of the sympathetic nervous system (Supplementary Fig. 16). To investigate whether the photothermal effect induced by PtNP-shell under NIR-II irradiation can precisely regulate the parasympathetic nerve, 100 μL PtNP-shell ($50 \, \mu g \, mL^{-1}$) and PBS were injected into NG of PtNP-shell group and control group (six beagle dogs in each group), respectively (Fig. 4a, b). It can be observed that upon irradiation with NIR-II laser ($0.8 \, W cm^{-2}$), the temperature of NG injected with PtNP-shell increased to 41.0 °C within a very short period of time (12 ± 3 s). Subsequently, the temperature of NG could be kept in the range of 41.0–42.9 °C for 5 min by adjusting the power density to $0.45 \, W cm^{-2}$ (Fig. 4c, d). As a crucial node within the parasympathetic neural network, activation of NG significantly reduces heart rate (HR) (Fig. 4e)[26]. Therefore, NG function was assessed by the maximum decrease in heart rate under direct electrical stimulation. As shown in Fig. 4f–h, NG function and activity were significantly elevated in the PtNP-shell group than in the control group after stimulation. The function and activity of NG recovered close to

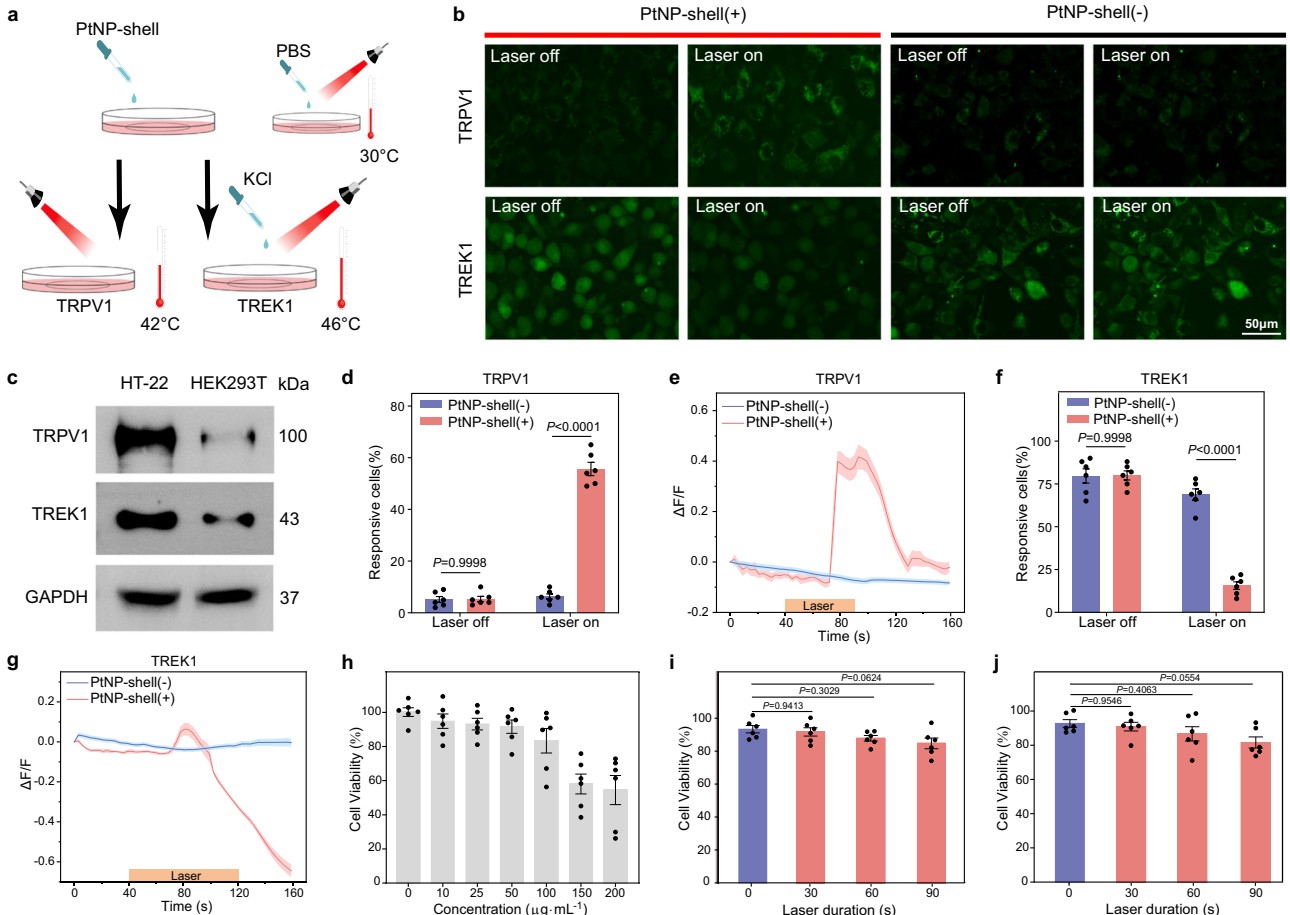

**Fig. 3 | PtNP-shell photothermal activation of different neuronal ion channels in vitro. a** Flowchart of calcium imaging assay performed on HT-22 cells. **b** calcium imaging of HT-22 cells under different experimental conditions, $n = 6$ biologically independent replicates. **c** Western blotting for TRPV1 and TREK1 from HT-22 and HEK-293T cells, $n = 4$ biologically independent replicates. Percentage of **d** TRPV1 and **f** TREK1 groups of HT-22 cells within the field of view of the fluorescence microscope that responded to laser stimulation, $n = 6$ biologically independent replicates. Temporal dynamics of Ca²⁺ signals in **e** TRPV1 and **g** TREK1 groups of cells. The solid lines indicate the mean, and the shade represents the standard error of the mean (SEM). **h** Cell viability of HT-22 treated with different concentrations of PtNP-shell for 24 h. Effect of NIR-II laser irradiation with varying durations on the viability of HT-22 cells treated with PtNP-shell (50 µg mL⁻¹) (Power densities: **i** 0.75 W cm⁻² and **j** 1 W cm⁻²), $n = 6$ biologically independent replicates. The error bar indicates S.E.M. Two-way ANOVA with Tukey's honestly significant difference (HSD) test was applied for statistical analysis of (**d**) and (**f**). One-way ANOVA with Dunnett's multiple comparisons test was applied for statistical analysis of (**i**) and (**j**). Source data are provided as a Source Data file.

baseline within 3 h after turning off the NIR-II laser, indicating that the photothermal modulation induced by PtNP-shell was reversible within NGs (Fig. 4h, Supplementary Figs. 17, 18). After NIR irradiation with the same parameters, the local temperature increase of the NG was <2 °C, while there was no significant change in parasympathetic nerve function (Supplementary Fig. 19).

In addition, the stability of the electrophysiology of the heart is reflected by measuring the effective refractory period (ERP) in various regions, including left ventricular apex (LVA), median left ventricular area (LVM), and left ventricular base (LVB). In the PtNP-shell group, the ERP was significantly elevated compared to the control group and remained elevated for 1 h after photothermal intervention in NG (Supplementary Fig. 20). Furthermore, immunofluorescence staining for vesicular acetylcholine transporter protein (VAChT), c-Fos, and TRPV1 on histopathological sections of photothermally modulated NGs served to localize parasympathetic neurons and reflect neuronal activity as well as TRPV1 protein expression, respectively (Fig. 4i). Quantitative analysis (Supplementary Fig. 21) revealed a substantial increase in the proportion of TRPV1⁺ (86.63 ± 2.65 vs. 45.45 ± 2.98) and c-Fos⁺ (77.81 ± 3.91 vs. 17.27 ± 3.08) neurons among VAChT⁺ parasympathetic neurons in the PtNP-shell group compared to the control group (all $P < 0.001$). These findings suggest that PtNP-shell can precisely regulate temperature and subsequently activate TRPV1 ion channels on NG to enhance parasympathetic activity.

## PtNP-shell photothermal activation of NG reduces I/R injury and associated VAs

Animal modeling and intervention manipulations were conducted to further elucidate the protective effects of precise modulation of NG by PtNP-shell against myocardial ischemia-reperfusion (I/R) injury and associated VAs, following the experimental protocols depicted in Fig. 5a, b. PtNP-shell and PBS were microinjected into the NG of the PtNP-shell group and control group, respectively, each consisting of six beagle dogs. The left anterior descending (LAD) coronary artery was occluded for 1 h to induce myocardial ischemia. Subsequently, NIR-II laser irradiation was applied to NG for 5 min, followed by reperfusion treatment achieved by opening the ligated knot.

Following I/R injury, electrocardiography (ECG) was recorded to monitor the occurrence of VAs events within 1 h, including ventricular premature beats (VPBs), ventricular tachycardia (VT), and ventricular fibrillation (VF) (Fig. 5c)[27]. Under NIR-II laser irradiation, the PtNP-shell group exhibited a lower incidence of sustained VT (sVT, duration >30 s) or VF compared to the control group (50% vs. 83%) (Fig. 5d). Moreover, the number of recorded VPBs (70.83 ± 5.38 vs.

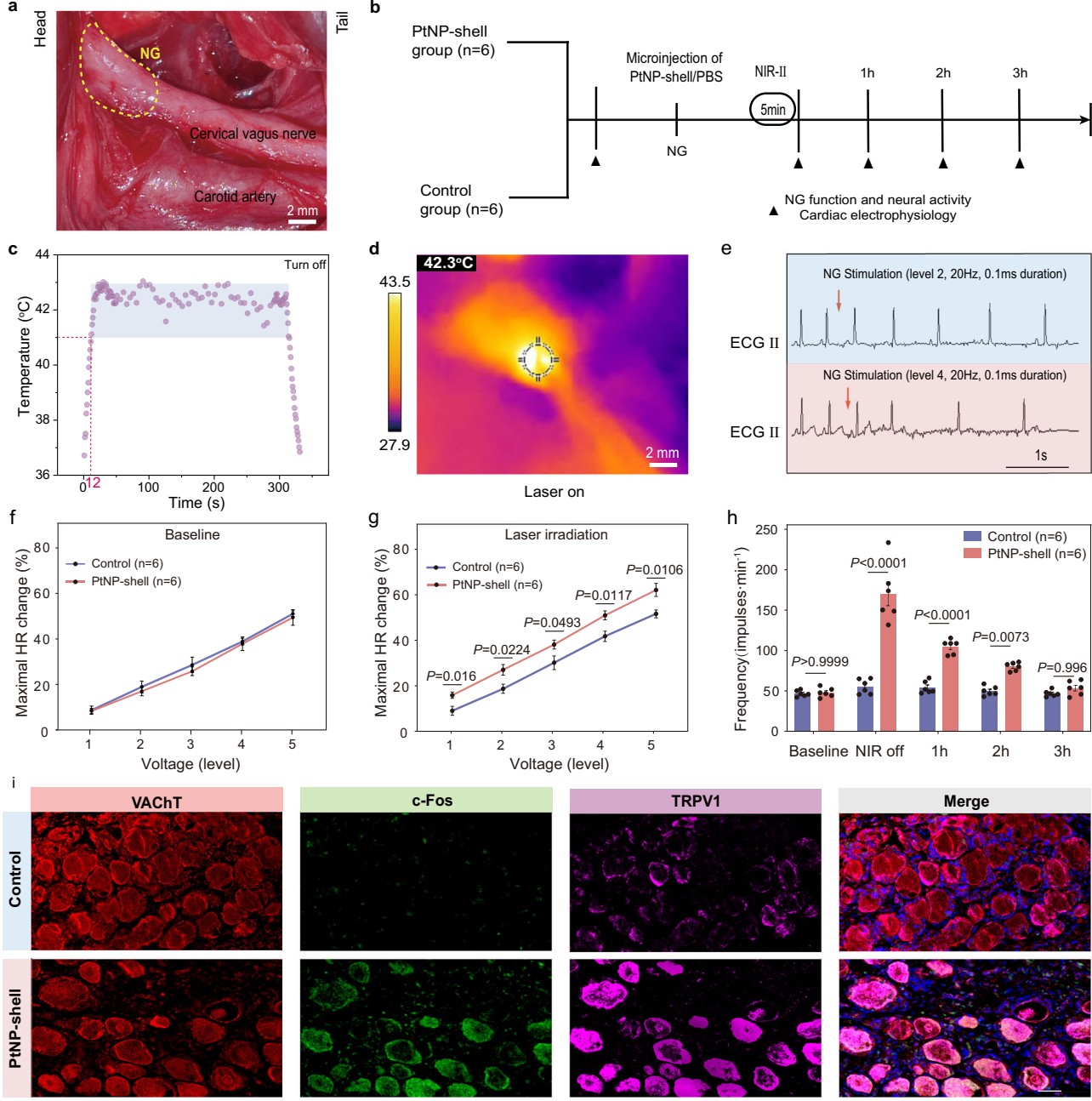

**Fig. 4 | Photothermal activation of the parasympathetic nervous system by PtNP-shell. a** Location of the canine NG. **b** Schematic illustration of the process of photothermal modulation of NG. **c** Temperature curves of NG under NIR-II laser irradiation. **d** Typical thermal imaging diagram of photothermally modulated activation of NG. **e** Representative images of HR reduction induced after stimulation of NG with different voltages. Maximal HR changes of beagle treatment with PtNP-shell or control **f** before and **g** after NIR-II exposure, *n* = 6 biologically independent replicates. **h** Quantification of the NG neural activity recordings, *n* = 6 biologically independent replicates. **i** Representative immunofluorescent images of VAChT, c-Fos, and TRPV1 in the NG of beagles following different treatments. Data are shown as the mean ± S.E.M. Unpaired two-tailed Student's *t*-test was applied for statistical analysis of (**g**). Two-way ANOVA with Tukey's HSD test was applied for statistical analysis of (**h**). NG nodose ganglion, HR heart rhythm. Source data are provided as a Source Data file.

116.00 ± 6.36, *P* < 0.05), VTs (3.17 ± 0.87 vs. 8.83 ± 2.15, *P* < 0.05) and duration of the VTs (7.00 ± 3.173 s vs. 26.83 ± 7.89 s, *P* < 0.05) in the PtNP-shell group were significantly reduced compared to that in the control group (Fig. 5e–g).

There were no statistically significant differences between the two groups in terms of preoperative ERP for LVB, LVM, and LVA. In the postoperative period, all three positions showed shortened ERPs in the control group. The PtNP-shell group exhibited significantly higher ERPs compared to the control group, indicating that photothermal modulation of nerves by PtNP-shell has a protective effect on cardiac

electrophysiology (Fig. 5h, i). Serum ELISA assay revealed significantly lower levels of markers of myocardial injury (MYO and c-TnI) at 4–5 h post-infarction in the PtNP-shell group compared to the control group (all *p* < 0.05, Fig. 5j, k), indicating an improvement in myocardial injury[28]. This may be attributed to the activation of α-7 nicotinic acetylcholine receptor by stimulating parasympathetic nerves, thereby alleviating inflammatory reactions and oxidative stress[29,30]. Postoperatively, heart rate variability analysis demonstrated lower low frequency (LF) and higher high frequency (HF) and the lower ratio of LF to HF (LF/HF) values in the PtNP-shell group compared to the

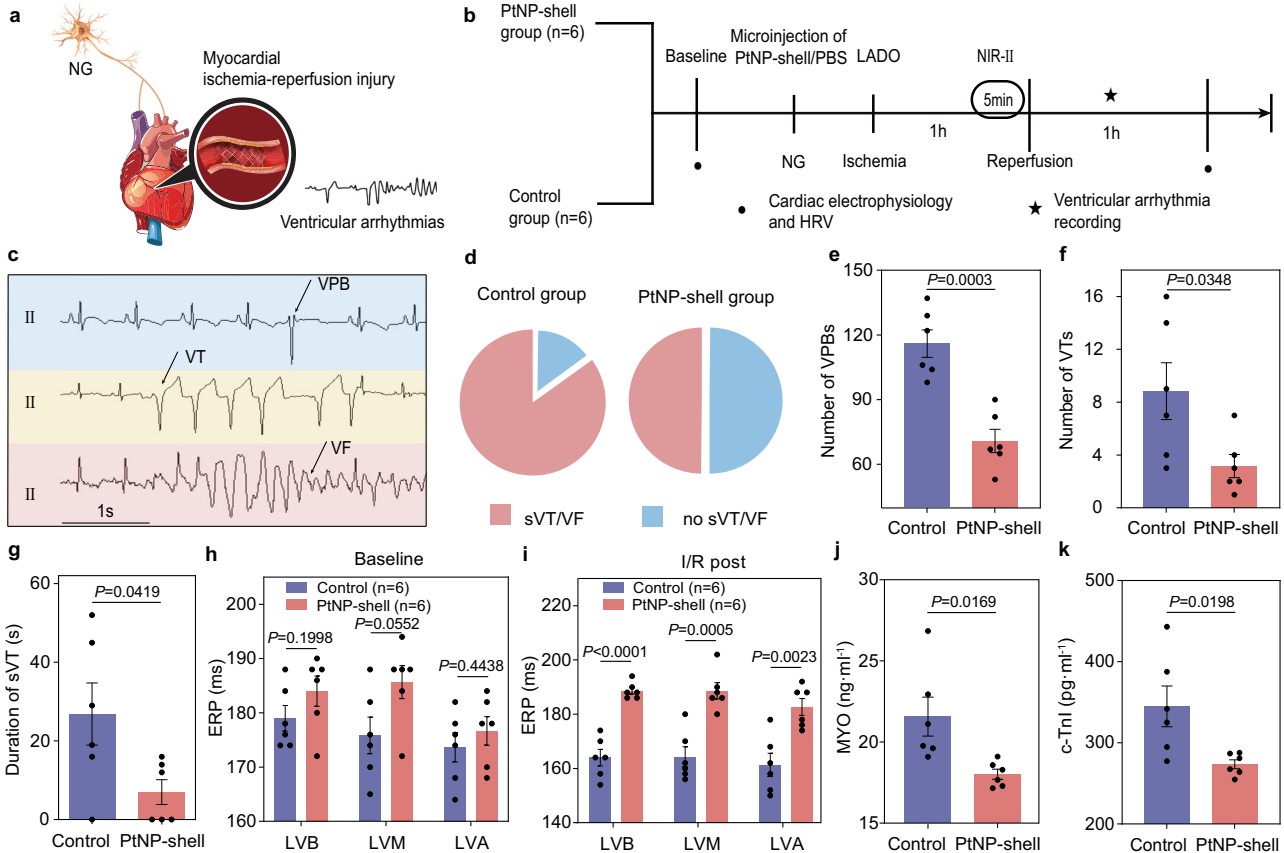

**Fig. 5 | PtNP-shell photothermal activation of the parasympathetic nervous system improves myocardial I/R injury. a** Schematic diagram and **b** flowchart of regulating NG to protect against myocardial I/R injury and associated VAs. **c** Representative visual depictions of VAs, including VPB, VT, and VF. **d** Quantitative analysis of the ratio of sVT and VF incidence between different groups, *n* = 6 biologically independent replicates. Quantitative analysis of the number of **e** VPBs, **f** VTs, and **g** the duration of sVT of beagles. Effects on ventricular ERP at different sites in beagles treatment with PtNP-shell or control **h** before and **i** after myocardial I/R injury modeling. Levels of markers of myocardial injury, including **j** MYO and **k** c-TnI, after different treatments in beagles *n* = 6 biologically independent replicates. Data are shown as the mean ± S.E.M. Unpaired two-tailed Student's *t*-test was applied for statistical analysis of (**e**–**k**). NG nodose ganglion, LADO left anterior descending coronary occlusion, VPB ventricular premature beat, VT ventricular tachycardia, sVT sustained VT, VF ventricular fibrillation, ERP effective refractory period, LVA left ventricular apex, LVM median left ventricular area, LVB left ventricular base. Source data are provided as a Source Data file.

control group (all *p* < 0.05, Supplementary Fig. 22). These results suggest that PtNP-shell reduces VAs by activating the parasympathetic nerve.

## PtNP-shell photothermal inhibition of the sympathetic nervous system

The sympathetic nervous system was modulated by performing microinjections of PtNP-shell or PBS into the LSG, followed by irradiation with a NIR-II laser (Fig. 6a, b). The temperature curve demonstrates that upon exposure to a NIR-II laser (0.8 W cm⁻¹) for 25 ± 5 s, the temperature rapidly escalated to 45.0 °C, crossing the range of 41.0–42.9 °C within a mere duration of 6 ± 1 s. Subsequently, the power density was immediately decreased to 0.6 W cm⁻², effectively maintaining LSG at a steady temperature between 45.0 and 46.9 °C (Fig. 6c, d). Due to the substantial increase in systolic blood pressure (SBP) induced by LSG activation (Fig. 6e), the function of LSG was evaluated by quantifying the maximum SBP change corresponding to five consecutive incremental voltages of high-frequency electrical stimulation. After 5 min of NIR-II laser irradiation, the activity and function of LSG in the PtNP-shell group were significantly suppressed compared to the control group (*p* < 0.05) and they returned close to baseline after 3 h (Fig. 6f–h and Supplementary Figs. 23, 24). Similarly, after NIR irradiation with the same parameters, the local temperature increase in the LSG did not exceed 2 °C, while there was no significant change in sympathetic nerve function (Supplementary Fig. 25).

Prolonged ERP effects were observed in all left ventricles, while the protective effect exhibited a duration of only 2 h (Supplementary Fig. 26). Furthermore, immunofluorescence staining was conducted on LSG tissues to examine the expression of tyrosine hydroxylase (TH), c-Fos, and TREK1 (Fig. 6i). These markers indicate the localization of sympathetic neurons, neuronal activity, and TREK1 protein expression, respectively. The quantitative analysis (Supplementary Fig. 27) revealed a significant decrease in the proportion of c-Fos⁺ expression in TH⁺ neurons within the PtNP-shell group (8.80 ± 1.80 vs. 44.78 ± 5.55, *P* < 0.001) indicating that PtNP-shell exerted a photothermal inhibitory effect on LSG neurons under NIR-II irradiation. However, the proportion of TREK⁺ expression was significantly increased within TH⁺ neurons in the PtNP-shell group (83.51 ± 3.72 vs. 57.20 ± 5.89, *P* < 0.01). This increase could lead to hyperpolarization of the cell membrane potential, reduction in neuronal excitability and inhibition of sympathetic nerve activity.

## PtNP-shell photothermal inhibition of LSG improves MI and reduces associated VAs

In order to investigate the impact of PtNP-shell photothermal effect on reducing VAs occurrence induced by MI, NIR-II light was applied to make LSG reach the target temperature of about 46.0 °C before ligating LAD coronary artery (Fig. 7a, b). Under NIR-II laser irradiation, the PtNP-shell group exhibited a significantly reduced incidence of sVTs or VF compared to the control group (16% vs. 50%) (Fig. 7c). In the PtNP-shell group, ECG recordings within infarction 1 exhibited a

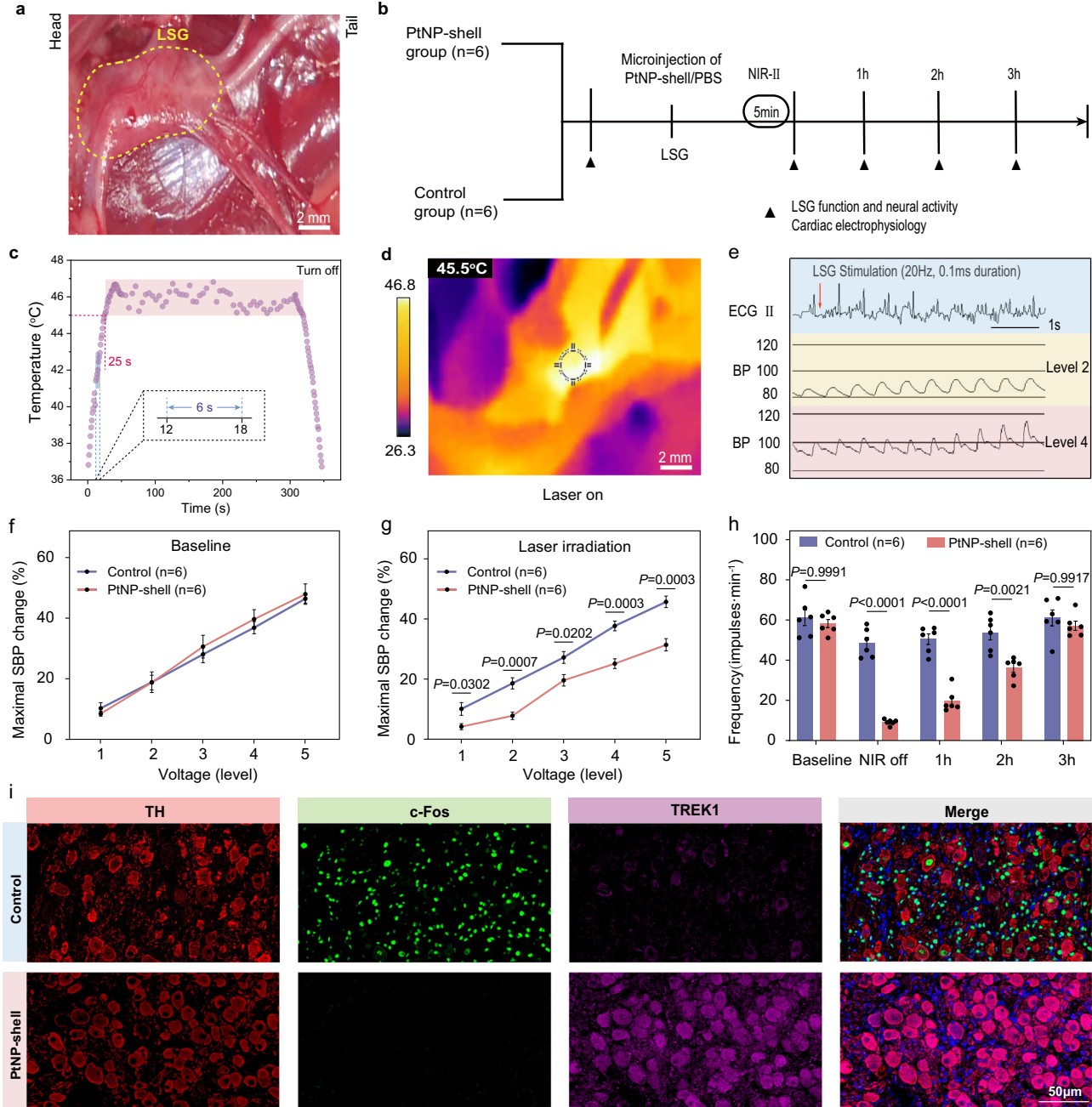

**Fig. 6 | Photothermal inhibition of the sympathetic nervous system by PtNP-shell. a** Location of the canine LSG. **b** Schematic illustration of the process of photothermal modulation of LSG. **c** Temperature curves of LSG under NIR-II laser irradiation. **d** Typical thermal imaging diagram of photothermally modulated activation of LSG. **e** Representative images of BP elevation induced after stimulation of LSG with different voltages. Maximal SBP changes of beagle treatment with PtNP-shell or control **f** before and **g** after NIR-II exposure, *n* = 6 biologically independent replicates. **h** Quantification of the LSG neural activity recordings, *n* = 6 biologically independent replicates. **i** Representative immunofluorescent images of TH, c-Fos, and TREK1 in the LSG of beagles following different treatments. Data are shown as the mean ± S.E.M. Unpaired two-tailed Student's *t*-test was applied for statistical analysis of (**g**). Two-way ANOVA with Tukey's HSD test was applied for statistical analysis of (**h**). LSG left stellate ganglion, BP blood pressure, SBP systolic BP. Source data are provided as a Source Data file.

reduced incidence of VAs events compared to the control group, with fewer VPBs recorded in the PtNP-shell group than in the control group (51.50 ± 5.53 vs. 70.83 ± 5.375, *P* < 0.05, Fig. 7d). However, there were no significant differences between the two groups in terms of VT numbers and duration (Supplementary Fig. 28). Additionally, VA inducibility measurements demonstrated that after photothermal neuromodulation with PtNP-shell, there was a decrease in VA score (1.50 ± 0.76 vs. 4.83 ± 1.14, *P* < 0.05) effective heart protection (Fig. 7e, f). Furthermore, PtNP-shell photothermal inhibition of LSG

produced similar protective effects on ventricular electrophysiological index ERP as activation of NG (Fig. 7g, h), and had a higher VF threshold than the control group (24.33 ± 4.24 vs. 12.33 ± 3.16, *P* < 0.05, Fig. 7i). In addition, the light inhibition of LSG followed the same trend as heart rate variability after activation of NG (Supplementary Fig. 29). These results suggest that PtNP-shell protects against cardiac damage and reduces VAs by modulating the autonomic nervous system, specifically by decreasing sympathetic activity and enhancing parasympathetic tone.

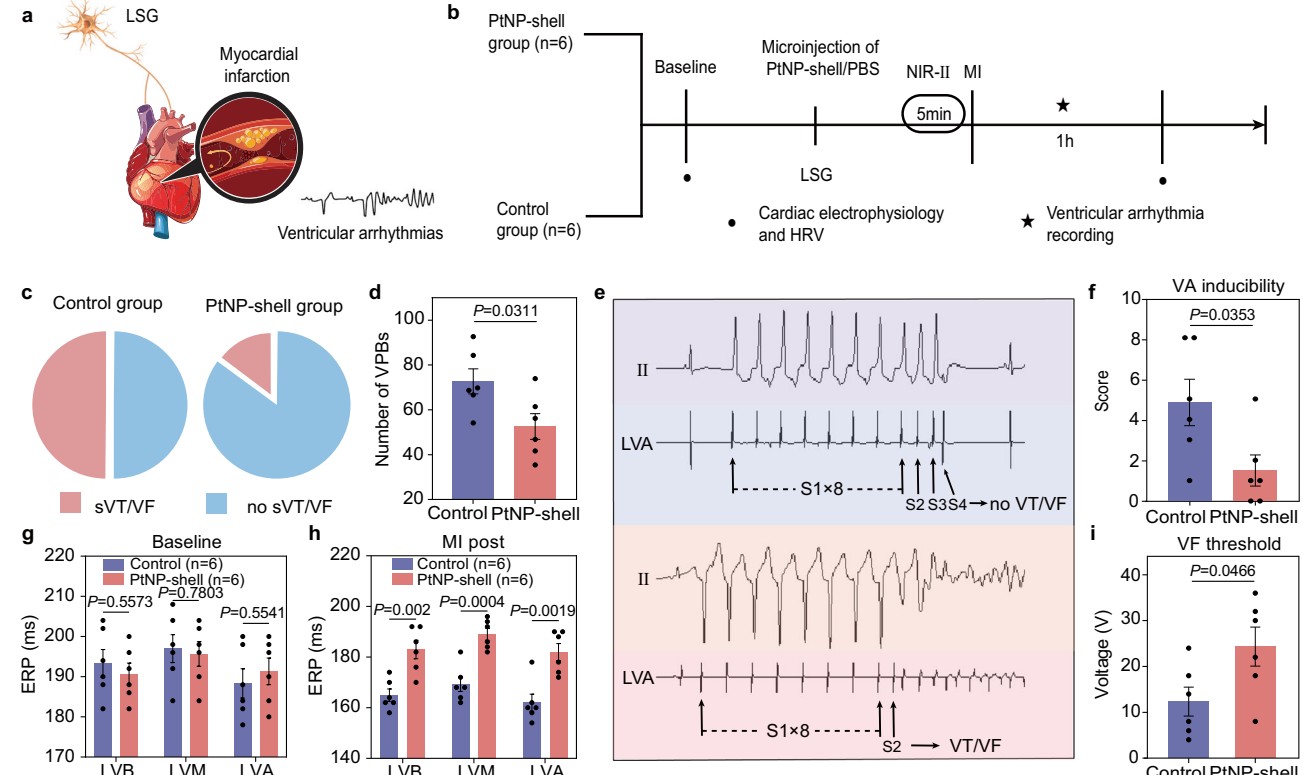

**Fig. 7 | PtNP-shell photothermal inhibition of the sympathetic nervous system improves MI-associated VAs.** Modulation of LSG to protect against MI and associated VAs **a** schematic diagram and **b** flowchart. **c** Quantitative analysis of the ratio of sVT and VF incidence between different groups, $n = 6$ biologically independent replicates. **d** Quantitative analysis of the number of VPBs of beagles. **e** Typical images of VA induced by programmed electrical stimulation. **f** Quantitative analysis of VAs score in different groups. Effects on ventricular ERP at different sites in Beagles treatment with PtNP-shell or control **g** before and **h** after MI modeling.

**i** Quantitative analysis of VF threshold in different groups, $n = 6$ biologically independent replicates. Data are shown as the mean ± S.E.M. Unpaired two-tailed Student's $t$-test was applied for statistical analysis of (**d**, **f**–**i**). LSG left stellate ganglion, SBP systolic blood pressure, MI myocardial infarction, VPB ventricular premature beat, VT ventricular tachycardia, VF ventricular fibrillation, VA ventricular arrhythmia, ERP effective refractory period, LVA left ventricular apex, LVM median left ventricular area, LVB left ventricular base. Source data are provided as a Source Data file.

## Biosafety of PtNP-shell for translational applications

To validate the biocompatibility of PtNP-shell photothermal modulation on the autonomic nervous system, we conducted rapid excision of LSG and NG tissues followed by hematoxylin and eosin (H&E) staining and Terminal deoxynucleotidyl transferase (TdT) dUTP Nick-End Labeling (TUNEL) assay. As shown in Supplementary Fig. 30, H&E and TUNEL staining did not reveal any indications of neuronal damage in both the PtNP-shell and control groups for both NG and LSG, indicating that the neuromodulation of PtNP-shell is repeatable. Meanwhile, to further investigate the long-term biosafety of PtNP-shell, a microinjection of 200 µl PtNP-shell (50 µg mL⁻¹) or PBS was administered into the ganglion of dogs and the tail vein of rats, respectively. After a follow-up period of 30 days, no significant damage was detected in the ganglia or major organs, including the heart, liver, spleen, lungs, and kidneys (Supplementary Figs. 30, 31a, i). Furthermore, blood biochemical analyses indicated no hepatotoxicity, nephrotoxicity, and induction of inflammatory responses (Supplementary Fig. 31b–h, j–p). These results unequivocally demonstrate that PtNP-shell exhibits exceptional biocompatibility and long-term biological safety.

## Discussion

The PtNP-shell reported in this study exhibits nearly perfect blackbody absorption properties, making it an efficient absorber with a high PCE in the NIR-II window (73.7% at 1064 nm). Furthermore, local heating induced by PtNP-shell activation effectively triggers temperature-sensitive ion channels TRPV1 and TREK1, enabling precise and efficient

modulation of autonomic nerves. This innovative approach holds great potential for non-invasive treatment of MI and associated VAs, as well as protection against reperfusion injury during interventional therapy.

Cardiac sympathetic denervation is a clinical procedure aimed at targeting the autonomic ganglia for refractory ventricular arrhythmias. However, ganglion removal can be traumatic for patients and may lead to complications due to the loss of original physiological function[31]. Currently, $\beta$-blockers are the primary pharmacological drugs employed in clinical practice for arrhythmia treatment[32,33]. However, their administration during acute myocardial ischemia remains unclear and is contraindicated in patients with heart failure. Additionally, previous research investigated the local ganglion blockade using botulinum toxin A to protect the heart[34]. Nevertheless, its prolonged blocking effect renders it unsuitable for acute myocardial ischemia management. Conversely, PtNP-shell-based photothermal neuromodulation offers reversible modulation within a short timeframe, exhibiting superior efficacy and controllability. In this study, we validated the protective efficacy of PtNP-shell photothermal neuromodulation strategy in models of acute myocardial infarction and acute reperfusion injury to mitigate ventricular arrhythmia incidence. However, further evaluation through experiments such as assessment of cardiac function and infarct area is required to determine the cardioprotective potential of this strategy in chronic myocardial injury models.

Moreover, the minimal tissue damage caused by light can be disregarded within the maximum permissible exposure (MPE) range,

rendering it one of the safest interventions for organisms. The interaction between light and tissue is intricate, and further research could aid in selecting more suitable wavelengths to achieve deeper penetration within the MPE range. Leveraging the nearly impeccable blackbody absorption of PtNP-shell and ultrasound-guided microinjection technology, remote and precise neuromodulation strategies can be developed, holding promise for non-invasive protection against MI and reperfusion injury-associated VAs. Simultaneously, exploiting the presence of blood vessels surrounding the ganglion presents an opportunity to minimize photon propagation within tissues. Consequently, photothermal modulation via NIR-II fibers in proximity to the ganglion through vascular routes during interventional therapy emerges as a promising avenue for direct clinical translation. The significance of this approach extends beyond VAs as it exhibits broad therapeutic prospects for chronic diseases like refractory hypertension[35] and stable atherosclerosis[36] due to the wide distribution of autonomic nerves and the universality of nerve regulation.

## Methods

### Chemicals
The gallium was purchased from Shanghai Minor Metals Co., Ltd. Anhydrous ethanol (≥99.7%) and KOH (AR) were purchased from Sinopharm Chemical Reagent Co., Ltd. $Na_2PtCl_6$ (98%) was purchased from Shanghai Aladdin Biochemical Technology Co., Ltd. mPEG-$SH_{5000}$ was purchased from Shanghai Macklin Biochemical Co., Ltd. STR-identified correct HT-22 cells or human embryonic kidney 293T (HEK-293T) cells were purchased with the corresponding specialized cell culture media (Procell, Wuhan, China). Rabbit monoclonal anti-TRPV1 antibody (Cat. No. A23386, Clone No. ARC57842) used for protein blotting was purchased from ABclonal (Wuhan, China). Mouse monoclonal anti-TREK1 antibody (Cat. No. sc-398449, Clone No. F-6) used in western blot and immunofluorescence staining was purchased from Santa Cruz Biotechnology (TX, USA). Rabbit monoclonal Anti-c-Fos (Cat. No. A24620, Clone No. ARC63309), rabbit polyclonal anti-VAChT (Cat. No. A16068), rabbit polyclonal anti-TH (Cat. No. A12756) and rabbit polyclonal Anti-TRPV1 (Cat. No. A8564) antibodies used in immunofluorescence staining were purchased from ABclonal. Glyceraldehyde 3-phosphate dehydrogenase (GAPDH) was purchased from Servicebio (Wuhan, China). Serum troponin I (c-TnI) and myoglobin (MYO) ELISA kits were purchased from Mlbio (Shanghai, China). Tumor necrosis factor alpha (TNF-α) and Interleukin 6 (IL-6) ELISA kits were purchased from Jianglaibio (Shanghai, China). 4,6-diamidino-2-phenylindole (DAPI) was purchased from Servicebio (Wuhan, China).

### Instruments
The morphology of PtNP-shell was characterized by an F200 transmission electron microscope (TEM) (JEOL, Japan) operated at 200 kV. STEM and HRTEM images were obtained by a JEM-ARM200CF (JEOL, Japan) at 200 kV. The EDX elemental mapping was carried out using the JEOL SDD-detector with two 100 mm² X-ray sensors. X-ray diffraction (XRD) patterns were performed on a SmartLab 9 kW X-ray powder diffractometer (Cu-target, 0.154 nm) (Rigaku, Japan). XPS measurements were carried out with an ESCALAB 250Xi spectrometer (Thermo Fisher Scientific, USA) under vacuum. Ultraviolet–visible–near-infrared light (UV–vis–NIR) absorption spectra were collected using a UV-3600 spectrophotometer (Shimadzu, Japan). Zeta potential (Z) and dynamic light scattering (DLS) were recorded using a Zetasizer Nano ZSP (Malvern Panalytical, UK). The fluorescence microscopy images of HT-22 cells were acquired by FV3000 Microscope (Olympus, Japan), excited with a 488 nm laser. TEM of HT-22 cells was characterized by HT7800 TEM (HITACHI, Japan). SEM of HT-22 cells was characterized by SU8100 SEM (HITACHI, Japan). Beagle's respiration is maintained by a WATO EX-20VET ventilator (Mindray, Shenzhen, China). ECG and blood pressure data were recorded by a Lead 7000 Computerized Laboratory System

(Jinjiang, Chengdu, China). NIR-II light at 1064 nm is generated by LWIRPD-1064-5F laser (Laserwave, Beijing, China). Thermal imaging was obtained by FLIR C2 thermal imager (FLIR, USA). High-frequency electrical stimulation was performed by a Grass stimulator (Astro-Med; West Warwick, RI, USA). The electrical signals of autonomic nerves are recorded by the Power Lab data acquisition system (AD Instruments, NSW, Australia) and visualized and analyzed by Labchart software (version 8.0, AD Instruments). Serum biochemical indices were determined by a fully automatic biochemical analyzer BK-1200 (BIO-BASE, Jinan, China).

### Synthesis of GaNPs
The GaNPs were obtained by sonication of liquid Ga. The liquid Ga (300 mg) was transferred to anhydrous ethanol (8 mL), and the solution was sonicated by nanoprobe sonication for 1 h (3 s on and 3 s off) at the power of 290 W. Then the ethanol was replaced with Milli-Q water to continue sonication for 1 h. The solution at the end of sonication was collected and centrifuged at $102 \times g$ for 5 min, and the upper liquid layer was aspirated for later use.

### Synthesis of PtNP-shell
First, the GaNPs and 3 mL $Na_2PtCl_6$ (0.1 M) were evacuated for 30 min and Ar was introduced for 15 min. Then, 3 mL $Na_2PtCl_6$ (0.1 M) was added dropwise to GaNPs and the solution was stirred for 4 h (droplet volume: 0.02 mL, dropwise rate: 1 droplet/s). After the reaction, the solution was collected and centrifuged at $8326 \times g$ for 10 min. The solids at the bottom were washed with Milli-Q water three times and finally dispersed in 6 mL Milli-Q for later use.

### Functionalization of PtNP-shell with mPEG-$SH_{5000}$
The PtNP-shell was first covered with a small amount of mPEG-SH to protect the structure from KOH. 30 mg mPEG-$SH_{5000}$ was added to 6 mL PtNP-shell and the solution was stirred for 12 h. After the reaction, the solution was collected and centrifuged at $8326 \times g$ for 10 min. The solids at the bottom were washed with Milli-Q water three times and dispersed in 6 mL Milli-Q water. The above solution was stirred with 12 mL of KOH (1 M) for 4 h. The reaction-completed solution was collected and centrifuged at $8326 \times g$ for 10 min, and the solids at the bottom were washed three times with Milli-Q water and finally dispersed in 6 mL Milli-Q water. The above solution was stirred with 60 mg mPEG-$SH_{5000}$ for 12 h. After the reaction, the solution was collected and centrifuged. The solids at the bottom were washed with Milli-Q water three times and finally dispersed in 6 mL PBS.

### Synthesis of Ga@Pt NPs
First, the GaNPs and 1 mL $Na_2PtCl_6$ (0.04 M) were evacuated for 30 min and Ar was introduced for 15 min. Then, 1 mL $Na_2PtCl_6$ (0.04 M) was added dropwise to GaNPs and the solution was stirred for 10 min (droplet volume: 0.02 mL, dropwise rate: 1 droplet/s). After the reaction, the solution was collected and centrifuged at $8326 \times g$ for 10 min, washed three times with Milli-Q water, and dispersed in 6 mL Milli-Q water. The above solution was stirred with 60 mg mPEG-$SH_{5000}$ for 12 h. After the reaction, the solution was collected and centrifuged ($8326 \times g$, 10 min). The solids at the bottom were washed with Milli-Q water three times and finally dispersed in 6 mL PBS.

### Calculation of the photothermal conversion efficiency
The relationship between temperature rise and energy transfer in the system can be described by Eq. 1,

$$\sum_i m_i c_i \frac{dT}{dt} = Q_{abs} - Q_{ext} = Q_{NPs} + Q_{solvent} - Q_{ext} \qquad (1)$$

where $Q_{abs}$ is the total energy absorbed by the system, $Q_{NPs}$ is the energy absorbed by the nanoparticles, $Q_{solvent}$ is the energy absorbed

by the solvent, $Q_{ext}$ is the energy loss from the system to the environment. $m_i$ and $c_i$ are the mass and specific heat capacity of the solution, respectively. $T$ is the solution temperature and $t$ is the irradiation time. The conversion of the light energy into heat energy can be expressed in terms of Eq. 2,

$$Q_{NPs} = I(1 - 10^{-A})\eta \qquad (2)$$

where $I$ is the laser power, $A$ is the absorbance value of PtNP-shell at 1064 nm, $\eta$ is the photothermal conversion efficiency. $Q_{solvent}$ can be calculated by the following Eq. 3,

$$Q_{solvent} = hs(T_{solvent} - T_{surr}) \qquad (3)$$

where $h$ is the convective heat transfer coefficient and $s$ is the surface area of the sample cell. $T_{solvent}$ is the maximum temperature that the solvent can reach under laser irradiation. $T_{surr}$ is the ambient temperature. $Q_{ext}$ can also be written as,

$$Q_{ext} = hs(T - T_{surr}) \qquad (4)$$

The heat output will increase with the increase in temperature when the NIR-II laser power is determined according to Eq. 4. The temperature of the system will reach the maximum when the heat input is equal to the heat output, so the following equation can be obtained,

$$Q_{NPs} + Q_{solvent} = Q_{ext-max} = hs(T_{max} - T_{surr}) \qquad (5)$$

where $Q_{ext-max}$ is the heat transferred from the system surface through the air when the sample cell reaches equilibrium temperature, and $T_{max}$ is the equilibrium temperature. Combining Eqs. 2, 3, and 5, $\eta$ can be expressed as,

$$\eta = \frac{hs(T_{max} - T_{surr}) - hs(T_{solvent} - T_{surr})}{I(1 - 10^{-A})} = \frac{hs(T_{max} - T_{solvent})}{I(1 - 10^{-A})} \qquad (6)$$

where $A$ is the PtNP-shell absorption at 1064 nm. To obtain $hs$, the dimensionless temperature $\theta$ is introduced,

$$\theta = \frac{T - T_{surr}}{T_{max} - T_{surr}} \qquad (7)$$

and a time constant of sample system, $\tau_s$

$$\tau_s = \frac{\sum_i m_i c_i}{hs} \qquad (8)$$

Combining Eqs. 1, 4, 7, and 8, the following equation can be obtained,

$$\frac{d\theta}{dt} = \frac{1}{\tau_s}\left[\frac{Q_{NPs} + Q_{solvent}}{hs(T_{max} - T_{surr})} - \theta\right] \qquad (9)$$

After the laser is turned off, in the cooling stage, there is no external input energy, $Q_{NPs} + Q_{solvent} = 0$, and Eq. 9 can be written as,

$$dt = -\tau_s \frac{d\theta}{\theta} \qquad (10)$$

By integrating Eq. 10, the following equation can be obtained,

$$t = -\tau_s \ln\theta \qquad (11)$$

Therefore, the system heat transfer time constant ($\tau_s$) at 1064 nm is 242.25 s (Fig. 3f). In addition, $m$ is 1 g and $c$ is 4.2 J g$^{-1}$. Therefore, $hs$

can be determined from Eq. 8. The laser power ($I$) used here can be determined as 1 W. Then the photothermal conversion efficiency ($\eta$) of the PtNP-shell at 1064 nm can be calculated to be 73.7% by substituting $hs$ into Eq. 6.

## Animal preparation and cell culture
All animal experiments were approved by the Animal Care and Use Committee of Renmin Hospital of Wuhan University (WDRM20230805A). All experimental procedures were in accordance with the Declaration of Helsinki for animals and were conducted according to the guidelines established by the National Institutes of Health. All adult male beagle dogs (8–12 kg) were anesthetized intravenously with 3% sodium pentobarbital (30 mg kg$^{-1}$ induction dose, 2 mg kg$^{-1}$ maintenance dose/h) and respiration was maintained by endotracheal intubation using a ventilator. Arterial blood pressure was continuously monitored through femoral artery catheterization with a pressure transducer attached. ECG and blood pressure data were recorded throughout the procedure. A heating pad was used to maintain the core body temperature at 36.5 ± 0.5 °C.

The cells were cultured in a humid incubator containing 5% CO$_2$ at a temperature of 37.0 °C. The cell-specific medium was prepared by mixing Dulbecco's modified Eagle's medium (DMEM), fetal bovine serum, and penicillin–streptomycin mixture at 89%, 10%, and 1%, respectively.

We performed separate experiments for each biological sample in our in vitro and in vivo experiments, performing duplicate experiments with different subjects to ensure the reproducibility and accuracy of our studies.

## Detection of TRPV1 and TREK1 expression in vitro and in vivo
Western blotting was used to assess the expression of TRPV1 and TREK1 in neuronal cells and ganglion tissues. HT-22 cells or HEK-293T cells were cultured in six-well plates for 24–48 h, then lysed and centrifuged (12000 × g, 10 min) to collect cells. Ganglion tissues were obtained from deceased animals and frozen in liquid nitrogen or stored at −80.0 °C. Total protein was determined using BCA protein assay reagent after tissue grinded and cells lysed. Afterward, the procedure was followed according to the manufacturer's instructions. Primary antibodies were anti-TRPV1 (dilution ratio 1:1000) and anti-TREK1 (dilution ratio 1:1000). Expression levels of specific proteins were normalized to GAPDH (dilution ratio 1:20000).

## Calcium imaging of neuronal cells
The effect of PtNP-shell photothermal modulation on ion channels in HT-22 cells was explored through calcium imaging experiments. HT-22 cells were incubated in 35 mm confocal dishes for 24 h. Cells were washed three times with PBS and then stained with 5 µM Fluo-4 AM (dilution ratio 1:500) for 30 min in a cell incubator at 37.0 °C, protected from light. To induce activation of TRPV1 and TREK1 ion channels, the culture dish was exposed to a 1064 nm laser (0.75 W cm$^{-2}$, TRPV1: 50 s, TREK1: 80 s), resulting in an elevation of temperature. TRPV1, being a calcium channel, exhibited observable changes in the flow of calcium ions upon activation, while TREK1 as a potassium channel did not display such behavior. Therefore, the effect of PtNP-shell photothermal modulation on neuronal cells via TREK1 was observed by introducing a 15 mM KCl solution prior to NIR-II irradiation. Fluorescence signals at 525 nm were recorded using a confocal microscope with 488 nm as the excitation wavelength. XYT images in the region of 1064 nm illumination were acquired and collected under a 20× objective lens. The average fluorescence intensity of the cells was analyzed using ImageJ software (Fiji). The normalized fluorescence change was calculated as follows: ΔF/F = (F − F$_0$)/F$_0$, where F is the original fluorescence signal; F$_0$ is the average baseline intensity before irradiation with NIR-II laser.

### In vitro cytotoxicity assay

The cytotoxicity of PtNP-shell on neuronal cells was evaluated by Cell Counting Kit-8 (CCK-8) assay. HT-22 cells were seeded in 96-well plates at a density of $1 \times 10^4$ well$^{-1}$ and cultured for 24 h. HT-22 cells were then treated with different concentrations (10, 25, 50, 100, 150, 200 µg mL$^{-1}$) of PtNP-shell for another 24 h. Cell viability was determined by CCK-8 assay after incubating with the CCK-8 reagent for 1 h. To investigate the impact of PtNP-shell's photothermal effect on neuron cell viability, HT-22 cells were co-cultured with PtNP-shell (50 µg mL$^{-1}$) for 12 h followed by irradiation with a 1064 nm laser (0.75 and 1 W cm$^{-2}$) for various durations (0, 30, 60, and 90 s). After incubation again for 12 h, the absorbance at 450 nm was recorded using a microplate reader. Cell survival (%) = (OD$_{samples}$ − OD$_{blank}$)/(OD$_{control}$ − OD$_{blank}$) × 100%.

### Experimental Protocol 1: activation of the parasympathetic nervous system through PtNP-shell photothermal reduces I/R injury

Part 1: exploring the in vivo effects of precise photothermal stimulation of the parasympathetic nervous system by PtNP-shell under NIR-II irradiation. Twelve beagles were randomly assigned to the control group (100 µL phosphate-buffered saline (PBS) was microinjected into the NG, $n = 6$) and the PtNP-shell group (100 µL PtNP-shell (50 g mL$^{-1}$) was microinjected into the NG, $n = 6$). NG nerve activity, heart rate (HR), and ventricular electrophysiological parameters were recorded at baseline and at multiple consecutive time points after NIR-II irradiation (Fig. 4b).

Part 2: the protective effect of PtNP-shell activation of the parasympathetic nervous system against myocardial I/R injury was investigated. The same grouping pattern as in Part 1 was used, with NIR-II irradiation (heating stage: 0.8 W cm$^{-2}$, 12 ± 3 s; equilibrium stage: 0.45 W cm$^{-2}$, 5 min) of the NG before opening the occluded LAD coronary vessel. Afterward, ventricular electrophysiological parameters, heart rate variability (HRV), and ECG data were recorded and analyzed (Fig. 5b).

### Experimental Protocol 2: PtNP-shell photothermal inhibition of the sympathetic nervous system improves MI

Part 1: the in vivo effects of precise photothermal stimulation of the sympathetic nervous system by PtNP-shell under NIR-II irradiation (heating stage: 0.8 W cm$^{-2}$, 25 ± 5 s; equilibrium stage: 0.6 W cm$^{-2}$, 5 min) were explored. Twelve beagles were randomly assigned to the control group (100 µL PBS microinjected into the LSG, $n = 6$) and the PtNP-shell group (100 µL PtNP-shell (50 g mL$^{-1}$) microinjected into the LSG, $n = 6$). LSG nerve activity, SBP, and ventricular electrophysiological parameters were recorded at baseline and at multiple consecutive time points after NIR-II irradiation (Fig. 6b).

Part 2: to investigate the protective effect of PtNP-shell inhibition on the sympathetic nervous system in improving MI, we used the same grouping pattern as in Part 1, with 5-min NIR-II irradiation of the LSG before ligation of LAD vessels. Finally, ventricular electrophysiological parameters, HRV, and ECG data were also recorded and analyzed (Fig. 7b).

### PtNP-shell photothermal stimulation of the autonomic nervous system in vivo

We selected NG and LSG as targets for modulation in the autonomic nervous system to explore the multimodality of the PtNP-shell photothermal strategy. A "C" incision is made behind the left ear, and the angle between the occlusal and trapezius muscles serves as the access approach[37]. The tissue is bluntly separated to expose the carotid sheath and identify the parasympathetic nerve. Moving upstream along the nerve, a distal expansion is observed as NG (Fig. 4a). LSG can be visualized and localized by left-sided thoracotomy (Fig. 6a)[34]. PtNP-shell (50 µg mL$^{-1}$) or PBS was slowly injected into two sites within the NG and LSG tissues to achieve homogeneous photothermal conversion. Initial vertical irradiation of NIR-II laser (1064 nm) at 0.80 W cm$^{-2}$

was performed on NG and LSG surfaces. The power density of the NIR-II laser was reduced to 0.45 W cm$^{-2}$ for continuous irradiation when the temperature of the NG reached 42.0 °C, and was reduced to 0.6 W cm$^{-2}$ for continuous irradiation when the temperature of the LSG reached 46.0 °C. The NIR-II laser irradiation remains stable with a spot size maintained at 1.0 cm$^{-2}$. Dual temperature monitoring using a thermal imager and T-type thermocouple was performed to plot the temperature-time curve.

### Functional assessment of autonomic nerves

The NG is a ganglion located upstream of the cervical parasympathetic nerve and can significantly inhibit HR after receiving direct electrical stimulation[26]. The LSG, as an important peripheral sympathetic ganglion, can rapidly elevate blood pressure when activated by electrical stimulation. Based on the functional properties of different autonomic ganglia, we assessed the function of NG and LSG. A pair of special electrodes made with silver wires were directly connected to the surfaces of NG and LSG for stimulation. High-frequency electrical stimulation (HFS: 20 Hz, 0.1 ms) was applied to the ganglion. The voltage was set to five levels in continuous increments (level 1: 0–2 V; level 2: 2–4 V; level 3: 4–6 V; level 4: 6–8 V; level 5: 8–10 V), while keeping the stimulation voltage values consistent with the baseline at different time points during the experiment. The percentage of sinus rate or AV conduction (measured by the A–H interval) slowing down constructed voltage level/degree of HR decrease curves reflecting NG function. On the other hand, the percentage increase in SBP built the voltage level/degree of SBP increase to reflect LSG function.

### Activity testing of autonomic nerves

Two specially designed microelectrodes were inserted into the NG and LSG, respectively, while a grounding wire was connected to obtain signals from the autonomic nerves. These electrical signals were recorded by a Power Lab data acquisition system, filtered through a band-pass filter (300–1000 Hz) and amplified 30–50 times by an amplifier. Finally, the signals were digitized and analyzed in LabChart software (version 8.0, AD Instruments).

### Construction of myocardial I/R injury model and MI model

The left anterior descending coronary occlusion (LADO) method was used to establish the MI model[27]. The ligation site was located beneath the first diagonal of the LAD, and the successful MI model was confirmed by observing ST-segment elevation on the ECG. After ensuring cardiac electrophysiological stabilization, the junction was released to reperfuse the occluded coronary arteries, completing the construction of the myocardial I/R injury model[38].

### Ventricular electrophysiological study in vivo

The ERP was measured at three locations: LVA, LVB, and LVM (located between the LVA and LVB). Malignant arrhythmic events occurring within 1 h of MI and I/R injury were assessed by electrocardiographic recordings in a canine model using a Lead 7000 Computerized Laboratory System. VAs were classified according to Lambeth Conventions as VPBs, VT (three or more consecutive VPBs), and VF[39]. The sVT is defined as continuous VT for more than 30 s. In addition, arrhythmia inducibility was further assessed by programmed ventricular stimulation at the right ventricular apex (RVA). Eight consecutive stimuli (S1S1) were performed at intervals of 330 ms, followed by additional stimuli until VT/VF occurred. Arrhythmia inducibility was assessed based on a modified arrhythmia scoring system[40]. If VF occurs during the evaluation, a defibrillator is required to restore sinus rhythm, followed by a waiting period of 30 min to restore cardiac electrophysiological stability. The VF threshold was assessed in the perimyocardial infarction region. Pacing was initiated using a Grass stimulator with a voltage of 2 V (20 Hz, 0.1 ms duration, 10 s). The stimulation voltage was increased in 2 V increments until VF was

induced. The lowest voltage that induced VF was regarded as the VF threshold[41].

## HRV analysis

The ECG data were recorded using the Power Lab data acquisition system. And the ECG segments recorded more than 5 min before modulation and after MI or I/R injury were analyzed by LabChart software with the Lomb−Scargle periodogram algorithm[42]. Frequency domain metrics of HRV were calculated, including LF (0.04−0.15 Hz, reflecting sympathetic tone), HF (0.15−0.4 Hz, reflecting parasympathetic tone), and LF/HF (reflecting autonomic balance). The results were expressed in standardized units.

## Immunofluorescence staining of histopathological sections

The ganglions were rapidly dissected for histopathological staining after the experimental animals died. Tissues were fixed with 4% paraformaldehyde, embedded in paraffin, and cut into 5 μm-thick sections. NG was stained with multiple immunofluorescence staining using anti-VAChT (dilution ratio 1:100), anti-c-Fos (dilution ratio 1:100), and anti-TRPV1 (dilution ratio 1:100) antibodies. VAChT is an essential protein for the transport of acetylcholine and can therefore localize parasympathetic neurons belonging to cholinergic neurons. Expression of c-Fos serves as a marker for neural cell activation. And LSG was stained by multiple immunofluorescences using anti-TH (dilution ratio 1:100), anti-c-fos (dilution ratio 1:200), and anti-TREK1 (dilution ratio 1:100) antibodies. Sympathetic neurons can be localized by TH protein expression. Cell nuclei were stained with DAPI. Images were taken at 100× magnification and analyzed using ImageJ software (Fiji).

## Enzyme-linked immunosorbent assay (ELISA)

In myocardial I/R injury model experiments, 5 mL of venous blood was obtained from the jugular vein of each beagle 4−5 h after ligation of the coronary vessels (3−4 h after reperfusion treatment). After standing for 1 h, the blood was centrifuged at $1237 \times g$ for 15 min. The upper serum layer was collected and stored at −80.0 °C. Myocardial injury levels were detected by c-TnI and MYO. Standard process analyses were performed according to the instructions of each ELISA kit. To evaluate the long-term biosafety and biocompatibility of PtNP-shell in vivo, beagle dogs and rats were randomly divided into PtNP-shell and PBS groups.

## Long-term biosafety assay in vivo

To evaluate the long-term biosafety and biocompatibility of PtNP-shell in vivo, beagle dogs and rats were randomly divided into two groups: a PtNP-shell group ($n = 6$) and a PBS group ($n = 6$). In the PtNP-shell group, 200 μL PtNP-shell (50 μg mL$^{-1}$) was microinjected into canine ganglion tissue and tail vein of rats to explore long-term biosafety. Blood and tissue samples were collected from each dog and rat 1 month after injection. One month after injection, blood samples were collected from the jugular vein of dogs as well as from the inferior vena cava of rats for analysis of serum biochemical indices, including alanine transaminase (ALT), aspartate aminotransferase (AST), urea, creatinine (Crea) and lactate dehydrogenase-1 (LDH1). Tissue H&E staining was also performed on major organs, including the heart, liver, spleen, lung, and kidney.

## Statistical analysis

All graphical data are presented as mean ± standard error of the mean (S.E.M.), and the distribution of data was assessed by the Shapiro−Wilk test. Differences between groups were determined using the Student's *t*-test or Mann−Whitney *U* test. Data distribution was assessed by the Shapiro−Wilk test for normality. Differences between the two means were tested using unpaired Student's *t*-test for Gaussian distributed data. Analysis of variance followed by one-way ANOVA with Dunnett test (one independent variable) or two-way ANOVA with Tukey test (two independent variables) were used as appropriate and indicated in the figure legend. The *P* values of <0.05 were considered statistically significant.

Data were analyzed and plotted using GraphPad Prism 9.0 software (GraphPad Software, Inc., La Jolla, CA, USA) and Oringinpro 2018 (OriginLab, USA).

## Reporting summary

Further information on research design is available in the Nature Portfolio Reporting Summary linked to this article.

## Data availability

All data underlying this study are available from the corresponding author upon request. Source data are provided with this paper.

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

## Acknowledgements

The research was supported by the National Natural Science Foundation of China (Grants 22025303: L.F., 82241057: L.L.Y., 82270532: L.L.Y., and 82200556: L.P.Z.); and the National Key Research and Development Program of China (Grant 2023YFC2705705: L.F.); and Foundation for Innovative Research Groups of Natural Science Foundation of Hubei Province, China (Grant 2021CFA010: L.L.Y.); and the Interdisciplinary Innovative Talents Foundation from Renmin Hospital of Wuhan University (Grant JCRCFZ-2022-005: H.J.). We thank the Core Facility of Wuhan University for their substantial support in sample characterization, including SEM, XPS, DLS, and XRD. We thank the Center for Electron Microscopy at Wuhan University for their support of STEM, HRTEM, and EDX characterization. We also thank Meimei Zhang in the Institute for Advanced Studies of Wuhan University for their assistance in TEM characterization.

## Author contributions

L.F., L.L.Y., and M.Q.Z. conceived the research concept. L.F., L.L.Y., M.Q.Z., and X.Y.Z. supervised the research; C.L.W., L.P.Z., C.Z.L., J.M.Q., X.R.H., B.X., Q.F.Q., Z.Z.Z., and J.L.W. performed the experiments; C.L.W., L.P.Z., C.Z.L., L.Y.W., and Y.X.L. discussed the results; C.L.W., L.P.Z., and C.Z.L. analyzed the data and co-wrote the manuscript. All authors commented on the manuscript.

## Competing interests

The authors declare no competing interests.
