## [Peer Review File · Nature Communications]

Reviewers' Comments:

Reviewer #1:

Remarks to the Author:

In this manuscript, the authors synthesized Pt nanoshells for cardiac protection through photothermal therapy. The photothermal conversion efficiency of the Pt nanoshells was studied. In the in vitro and in vivo studies, the effect of photothermal therapy on TRPV1 and TREK1 channels were studied. I have the following comments.

1. The effect of photothermal therapy in the NIR-I window on ventricular arrhythmias was reported by the authors previously. In this study, the authors used the laser irradiation in the NIR-II instead. In fact, the irradiation was directly performed at the NG and LSG with microinjection of the nanoparticles. There is no need to use PTT in the NIR-II. Therefore, this work lacks novelty.
2. In the supporting information, the authors suggested that the photothermal conversion efficiency of Pt nanoshells was higher than other nanoparticles in previous reports. It is not accurate.
3. The Pt nanoshells don't have a unique strong absorption peak in the NIR-II range. Therefore, they are not good candidates for PTT in the NIR-II window.
4. In the in vitro cell study, the PTT study was performed in a 35 mm confocal dish. However, the irradiation diameter of the laser was very small. Most the cells in the dish cannot received the laser irradiation. It should be another concern.

Reviewer #2:

Remarks to the Author:

Wang, Zhou and Liu present a paper describing a ~200 nm platinum-based, PEG-modified nanoparticle (NP) with high absorbance of near infra-red (1,000 nm+) light. The synthesised NP has high photothermal efficiency, which the authors used to stimulate heat-sensitive TRPV1 and TREK1 ion channels. First, they demonstrate this using HT-22 hippocampus neuronal cells, they then applied it to a canine myocardial infarction model, stimulating channels in the nodose ganglion (NG) and left stellate ganglion (LSG). The aim is to stimulate the parasympathetic nervous system and inhibit the sympathetic nervous system. The authors then show changes in ventricular electrophysiology and reduced biomarkers of cardiac injury, plus some basic biocompatibility data.

Overall, there are many positive aspects of the work. I find that the paper is very well presented, with nicely laid out, attractive and well designed figures with clear illustrations. The methods section, at the end of the main document, is also very detailed. However, I find that the main manuscript is quite difficult to follow in some places. There are a lot of abbreviations - many are used without first defining them (though they can be found in the methods section). There is also very little introduction or background information given, meaning that the rationale of the study is not very clearly spelled out. No hypothesis or research aims are stated.

I think it is also somewhat unusual that the introduction section starts to describe results (line 54 onwards) and even refers to figures (even if they are just schematic diagrams.)

Similarly, the results section offers very little narrative or explanation of the results, which makes it difficult to follow the rationale of each experiment. For example, Vacht and c-fos staining are introduced with no explanation of what those are, why they are being stained, or what those results actually mean.

The work appears to be original. The authors have previously used the same canine I/R models and stimulation of the NG and LSG by other means, but I do not find this NP formulation described previously. Statistical tests, sample sizes and P values are mostly clearly annotated and described, and seem appropriate for the types of data being analysed. Sample sizes also seem reasonable, though it not always clear which data points are replicates, independent biological samples, or

actual experimental repeats.

I do not find any major flaws with the NP synthesis or characterisation aspects, though this is not my specialty. However, I do have a few questions about the canine I/R model:

The text mentions that "The NG was subsequently exposed to NIR-II laser irradiation for a duration of 5 minutes prior to occlusion of the left anterior descending (LAD) coronary artery for reperfusion therapy." This text, and the diagram in Figures 4B, 5B and 6B, explain that the treatment was done before the I/R was induced. I am curious about why the authors choose a pre-treatment experimental design, rather than initiating the treatment after reperfusion. Obviously investigating only pre-treatment greatly lowers the clinical/therapeutic relevance of the research.

Related to this, why do the authors think there is a reduction of serum troponin and myoglobin? (It is also not clear at what time point these samples were taken. Does this reflect acute cardio protection?) In my opinion, if the authors want to say that the system is cardioprotective (line 292) or protects against cardiac damage (line 350), additional metrics (echocardiography, infarct volume etc) would be required to support this claim.

Figure 4f and 4g are a little difficult to understand. The Y axis is different between the two graphs, but it seems that the effect on max HR is quite mild? I think it would make sense to put the baseline and laser irradiation groups on the same graph and statistically compare those too.

I am curious about the overall delivery efficiency, and biodistribution of the NPs and whether direct injection into the NG/LSG is clinically applicable.

I also note that the in vivo comparisons are simply nanoparticles vs. PBS. This can clearly demonstrate that the NPs have some activity; but how do they compare to other methods of stimulating the NG/LSG, or drugs which stimulate the sympathetic/parasympathetic nervous system? Without these sorts of comparison, it is difficult to put the significance of authors' findings into context. If the authors can demonstrate that their approach is better than other approaches, this could be a lot more convincing.

In terms of overall interest, I think is a good technical demonstration of clever system; but what is the real-world application? Could the authors envisage a way in which this technology can actually be applied to MI patients?

The abstract mentions that the NPs conferred protection against ventricular arrhythmias following MI. However, supplementary figure 25 seems to show that there was no difference in overall VA events.

Minor points:

Figure 3i is not very easy to read or understand.

Line 373, I think that more than a few blood tests and some organ histologically is required to make such a strong claim of "unequivocally demonstrate" long-term safety.

Supplementary figure 6, 13. I think these would be more readable as tables rather than bar charts.

Supplementary figure 7; this is quite a broad range of nanoparticle sizes. What makes up the smaller (100 nm) particles? Is there aggregation to produce larger particles?

Reviewer #3:

Remarks to the Author:

The manuscript describes photothermal neuromodulation via Pt nano-shell nanoparticles. Ga nanoparticles are used as a template for electrocoupling substitution-based synthesis of the Pt nano-shell. Using KOH wet etching, Ga core is etched and a Pt nano-shell structure is obtained. The rough surface topography of the Pt nano-shell structure allows the particles to exhibit high

optical absorbance. It is claimed that these particles have one of the highest photothermal energy conversion efficiencies. The photothermal conversion of optical irradiation is then utilized for stimulation of target cells and tissues via temperature activated ion channels- TRPV1 and TREK1. The potential application of such photothermal modulation technique in regulating cardiac pulsing is demonstrated with regards to protecting against acute ventricular arrhythmias.

However, the manuscript does not include proper controls to demonstrate that in-vivo photothermal modulation is achieved exclusively through the Pt nanoparticles. In addition, there are certain claims and results that need to be better corroborated to reach the scientific requirements of the journal. Therefore, I cannot recommend that this manuscript be accepted at Nature Communications in its current form.

1. Abstract: It is claimed that "the autonomic nervous system plays a pivotal role in the pathophysiology of cardiovascular diseases." This sentence is misleading since the dysregulation of the autonomic nervous system can contribute to cardiovascular diseases. However, it is not the primary contributor to the diseases, autonomic nervous system in fact regulates normal functioning of the cardiovascular system. (see: Purves D, Augustine GJ, Fitzpatrick D, LaMantia AS, McNamara JO, Williams SM. Autonomic regulation of cardiovascular function. *Neuroscience*. 2001;491-3 AND Gordan R, Gwathmey JK, Xie LH. Autonomic and endocrine control of cardiovascular function. *World journal of cardiology*. 2015 Apr 4;7(4):204.)

2. The authors claim that bi-directional reversible autonomic modulation is achieved via NIR-II photothermal modulation using Pt nano-shell nanoparticles. The manuscript presents uni-directional modulation where the target tissues are stimulated. Bi-directionality isn't demonstrated since in terms of neural interfaces, bi-directionality refers to the ability to record neural activity as well as stimulate (see: Song KI, Seo H, Seong D, Kim S, Yu KJ, Kim YC, Kim J, Kwon SJ, Han HS, Youn I, Lee H. Adaptive self-healing electronic epineurium for chronic bidirectional neural interfaces. *Nature communications*. 2020 Aug 21;11(1):4195. AND Hughes C, Herrera A, Gaunt R, Collinger J. Bidirectional brain-computer interfaces. *Handbook of clinical neurology*. 2020 Jan 1;168:163-81.).

3. It is unclear why the NIR-II range was utilized in this work. This is important for selecting the right materials, models, and experiments. (see: [nature.com/articles/s44222-023-00022-y](https://www.nature.com/articles/s44222-023-00022-y) AND [nature.com/articles/s41551-022-00862-w](https://www.nature.com/articles/s41551-022-00862-w)).

4. Instead of using terminologies like "nearly perfect blackbody absorption", the actual optical properties and metrics should be presented.

5. Figure 1 presents how the nanoparticles will interact with the biological systems, however it does not show how light pulses/irradiation will be delivered to the target tissues/sites. This should be discussed in the figure and the manuscript since it is important for clinical translation.

6. Adequate controls should be provided to better compare the physical properties of PtNP-shells. That is, please provide the optical absorbance of GaNPs, Pt coated GaNPs for Figure 2.d; similar controls should be provided for Figure 2.e (including the thermal transients of such the solvent under irradiation).

7. For the XPS characterization, a survey scan of presentative sample should be presented along with the detailed XPS characterization of oxygen (O1s) and potassium (K2p). The elemental composition of the PtNP-shells, Pt coated GaNPs, and GaNPs should be compared as well. This will better elucidate the composition of effectiveness of the synthesis protocols.

8. The stability of the Pt-nanoshell suspensions should be evaluated as a function of time. Do the nanoparticle aggregate over time? Will this be a concern when the Pt-nanoshells are injected into biological systems.

9. How does the addition of mPEG-SH5000 effect the photothermal properties of the nanoparticles?

10. Critical information from the methods section is missing- for example, details regarding the cell culture protocol and photothermal stimulation (such as power and pulse duration of optical irradiation are missing). How long was the ECG data recorded for? What were the exact stimulation conditions for all in-vivo experiments?

11. For the in-vitro experiments, are the Pt nanoparticles engulfed by the target cells or are they localized in the vicinity of the cell membrane?

12. Both in-vivo and in-vitro photothermal stimulation experiments require the cells' microenvironment to reach temperatures greater than 42°C. Does repeated photothermal stimulation using such high temperatures adversely affect cellular health by disrupting the cell membrane or trigger heat shock response?
13. The claim that Pt-NP shell does not induce significant damage to neurons under controlled NIR-II laser irradiation is incorrect since there is ~10% loss in cellular viability.
14. It will be recommended that the data presentation in Figure 3.i be changed since the details of the data are difficult to comprehend through a 3-D plot.
15. For the in-vivo photothermal stimulation experiments, can similar affects be achieved without the presence of the Pt-nanoshell particles? Figure 4.d presents high temperature gradients for the surrounding tissue as well. Stimulation using infra-red radiation has been demonstrated previously, see: doi.org/10.1364/OL.30.000504 and doi.org/10.1117/1.2121772.
16. The biosafety of Pt-nanoshell particles was evaluated after a rapid excision of the LSG and NG tissues. Can the authors comments on the long-term biosafety of the nanoparticles in passive (without photothermal stimulation) and active (with photothermal stimulation) states?
17. Page 2, line 21: Please include examples and appropriate references for "conventional international procedures for MI."
18. Page 4, line 16: Please change the word "encapsulated on" since Pt is not encapsulated on the surface of GaNPs. Pt is deposited onto of GaNP core then it encapsulates GaNP core.

Reply to the referees

To Referee #1

First of all, we really appreciate your constructive comments. We have made a point-by-point response to your comments and carefully revised the manuscript as you suggested. For your reference, please find our revisions marked in red color.

Comment 1: *In this manuscript, the authors synthesized Pt nanoshells for cardiac protection through photothermal therapy. The photothermal conversion efficiency of the Pt nanoshells was studied. In the in vitro and in vivo studies, the effect of photothermal therapy on TRPV1 and TREK1 channels were studied. I have the following comments. The effect of photothermal therapy in the NIR-I window on ventricular arrhythmias was reported by the authors previously. In this study, the authors used the laser irradiation in the NIR-II instead. In fact, the irradiation was directly performed at the NG and LSG with microinjection of the nanoparticles. There is no need to use PTT in the NIR-II. Therefore, this work lacks novelty.*

Author reply: Thank you for your comments. The focus of our research is to utilize PtNP-shell with near-perfect blackbody absorption and high photothermal conversion efficiency, enabling safer and more precise bidirectional deep neural modulation of the nodose ganglion (NG) and the left stellate ganglion (LSG). Moreover, compared to the first near-infrared (NIR-I, 650–900 nm) and visible window, the photons in the second near-infrared window (NIR-II, 900–1700 nm) exhibit reduced tissue scattering and absorption, thereby increasing the maximum allowable exposure (MPE) of biological tissues. This means photons within the NIR-II window exhibit significantly enhanced tissue penetration depths (up to 5–20 mm) (*Nat. Nanotech.* **2009**, *4*, 710; *Nat. Med.* **2012**, *18*, 1841; *Nat. Biomed. Eng.* **2017**, *1*, 0010). We developed PtNP-shell and validated its photothermal neuromodulation efficacy in the NIR-II window both *in vivo* and *in vitro*. Given its wavelength independence, further investigations may facilitate the selection of a more suitable laser for achieving deeper tissue penetration while adhering to the MPE range. The exceptional potential of PtNP-shell makes it highly promising for precise neural regulation in deeper tissue. Moreover, NG and LSG are pivotal nodes of the vagal loop and sympathetic loop, respectively. In contrast to previous studies, our approach not only achieves neural activity inhibition but also enables nerve activation for bidirectional reversible modulation. Furthermore, we substantiate the therapeutic efficacy of this strategy in diverse models of cardiac injury.

Comment 2: *In the supporting information, the authors suggested that the photothermal conversion efficiency of Pt nanoshells was higher than other nanoparticles in previous reports. It is not accurate.*

Author reply: Thank you for the comment. The PtNP-shell exhibits an exceptionally high photothermal conversion efficiency, surpassing the values reported in Supplementary Table 2 (Table R1). To ensure utmost rigor, we have revised the statement to “among the highest”.

Table R1 | Comparison of photothermal conversion efficiency.

	Photothermal conversion efficiency (%)
This work	73.70
PEDOT:ICG@PEG-GTA ¹	71.10
MINDS ²	71.00
PTG NPs ³	67.60
RBC@Cu _{2-x} SeNPs ⁴	67.20
AuDAg ₂ S ⁵	67.10
MAPSULES ⁶	67.00
Fe ₃ O ₄ @PPy@GOD NCs ⁷	66.40
NPPBTPBF-BT ⁸	66.40
AS1064 ⁹	65.92
Gold Nanoraspberry ¹⁰	65.00
P-Pc-HSA ¹¹	64.70
Ultrathin polypyrrole nanosheets ¹²	64.60
H _x MoO ₃ ¹³	60.90
NiP PHNPs ¹⁴	56.80
FP NRs ¹⁵	56.60
COF ¹⁶	55.20
2MPT ⁺ -CB ¹⁷	54.60
SPN-PT ¹⁸	53.00
Pdots ¹⁹	53.00
Pt Spirals ²⁰	52.50
TBDOPV-DT ²¹	50.50
Ti ₂ O ₃ @HA NPs ²²	50.20
TBDOPV-DT NP ²³	50.00
DPP-IID-FA NPs ²⁴	49.50
SPN-DT ¹⁸	49.00
NP ²⁵	49.00
FTQ nanoparticles ²⁶	49.00
CNPs ²⁷	49.00
H-SiO _x NPs ²⁸	48.60
BETA NPs ²⁹	47.60
CN-NPs ³⁰	47.60
Pt-NDs ³¹	46.90
MoO _{3-x} nanobelts ³²	46.90

P3 NPs ³³	46.00
Ni ₉ S ₈ ³⁴	46.00
PtAg nanosheets ³⁵	45.70
Nb ₂ C NSs ³⁶	45.65
V ₂ C-TAT@Ex-RGD ³⁷	45.10
PEG-TONW NRs ³⁸	43.60
SPNI-II ³⁹	43.40
1T-MoS ₂ ⁴⁰	43.30
Bi@C NPs ⁴¹	43.20
Au NPL@TiO ₂ ⁴²	42.10
CT NPs ⁴³	42.00
PPy-PEG NPs ⁴⁴	41.97
AuPt@CuS NSs ⁴⁵	41.56
Bi ₁₉ S ₂₇ I ₃ nanorods ⁴⁶	41.50
Cu ₃ BiS ₃ NR ⁴⁷	40.70
MPAE-NPS ⁴⁸	40.07

Comment 3: *The Pt nanoshells don't have a unique strong absorption peak in the NIR-II range. Therefore, they are not good candidates for PTT in the NIR-II window.*

Author reply: The PtNP shell exhibits strong absorption across a broad range of wavelengths, including the NIR-II window. We posit that the PTNP-shell holds promise as an excellent candidate for photothermal therapy (PTT) within the NIR-II window. Given its wavelength independence, further investigations may facilitate selection of a more suitable laser for achieving deeper tissue penetration while adhering to the MPE range, thereby enabling precise and noninvasive neural modulation.

Fig. R1 | Absorption of PtNP-shell. **a**, Absorption curves of PtNP-shell with different concentrations (10, 25, 50 and 75 $\mu\text{g}\cdot\text{mL}^{-1}$). **b**, Mass extinction coefficient of PtNP-shell at 1064 nm. Normalized absorbance intensity at $\lambda = 1064$ nm divided by the characteristic length of the cell (A/L) at different concentrations (10, 25, 50 and 75 $\mu\text{g}\cdot\text{mL}^{-1}$).

Comment 4: *In the in vitro cell study, the PTT study was performed in a 35 mm confocal dish. However, the irradiation diameter of the laser was very small. Most the cells in the dish cannot received the laser irradiation. It should be another concern.*

Author reply: Thank you for the comment. With reference to previous photothermal research on cells (*Nat. Biomed. Eng.* **2022**, 6, 754; *Nano Converg.* **2022**, 9, 13), we conducted a cell calcium imaging experiment wherein the NIR-II laser was utilized to activate neuron cells within the confocal dish covered by the laser beam. We employed confocal microscopy to capture images of these neuron cells covered by laser beam and make quantitative analysis of calcium imaging. Consequently, the dimensions of the confocal dish did not influence the outcomes of the cellular calcium imaging experiment. A more comprehensive explanation is provided in the “Method” section:

“XYT images in the region of 1064 nm illumination were acquired and collected under a 20x objective lens.”

To Referee #2

We appreciate your positive evaluation on our manuscript. We have made a point-by-point response to your comments and carefully revised the manuscript as you suggested. For your reference, please find our revisions marked in red color.

Comment 1: *Wang, Zhou and Liu present a paper describing a ~200 nm platinum-based, PEG-modified nanoparticle (NP) with high absorbance of near infra-red (1,000 nm+) light. The synthesised NP has high photothermal efficiency, which the authors used to stimulate heat-sensitive TRPV1 and TREK1 ion channels. First, they demonstrate this using HT-22 hippocampus neuronal cells, they then applied it to a canine myocardial infarction model, stimulating channels in the nodose ganglion (NG) and left stellate ganglion (LSG). The aim is to stimulate the parasympathetic nervous system and inhibit the sympathetic nervous system. The authors then show changes in ventricular electrophysiology and reduced biomarkers of cardiac injury, plus some basic biocompatibility data. Overall, there are many positive aspects of the work. I find that the paper is very well presented, with nicely laid out, attractive and well designed figures with clear illustrations. The methods section, at the end of the main document, is also very detailed. However, I find that the main manuscript is quite difficult to follow in some places. There are a lot of abbreviations - many are used without first defining them (though they can be found in the methods section). There is also very little introduction or background information given, meaning that the rationale of the study is not very clearly spelled out. No hypothesis or research aims are stated.*

Author reply: Thank you for your recognition of our work. The advices you put forward are very constructive and help a lot for us to improve our manuscript.

We have provided reasonable descriptions of specialized acronyms that appear for the first time to increase readability. In addition, we have followed up on your comments to optimize the “introduction”. The background and rationale for the development and application of this strategy are described in more detail, and our research hypotheses and objectives are added.

“However, left cardiac sympathetic denervation (LCSD), stellate ganglion block (SGB), and renal denervation (RDN) are associated with certain adverse effects, including Horner's syndrome (*Circ.-Arrhythmia Electrophysiol.* **2015**, *8*, 1007), inadvertent bleeding, and inconsistent ablation outcomes (*N. Engl. J. Med.* **2014**, *370*, 1393). Both conventional low-level vagus nerve stimulation (LL-VNS) and optogenetic neuromodulation necessitate the implantation of *in vivo* electrical stimulation (*JACC-Clin. Electrophysiol.* **2017**, *3*, 929) or light source devices (*J. Am. Coll. Cardiol.* **2017**, *70*, 2778). Furthermore, optogenetic neuromodulation requires viral transfection of photosensitive proteins (*J. Am. Coll. Cardiol.* **2017**, *70*, 2778), thereby limiting the clinical advancement of these therapeutic approaches.”

“The light in the second near-infrared window (NIR-II, 900–1700 nm) has reduced tissue scattering and absorption and increased maximum permissible exposure (MPE) for biological tissues compared to the light in the first near-infrared (NIR-I, 650–900 nm) and visible window (Acc. Chem. Res. 2018, 51, 1840). Consequently, this enables non-invasive and non-implantable neuromodulation using NIR-II photothermal.”

“Therefore, the objective of this study is to develop a Pt nanoparticle shell (PtNP-shell) with near-blackbody properties and ultra-high PCE in the NIR-II window (Fig. 1a) and investigate its potential in protecting against MI and myocardial reperfusion injury accompanying intervention through rapid, efficient, and precise multifunctional autonomic neuromodulation (Fig. 1b).”

Comment 2: *I think it is also somewhat unusual that the introduction section starts to describe results (line 54 onwards) and even refers to figures (even if they are just schematic diagrams).*

Author reply: Following your suggestion, we have revised the introduction by modifying the section describing results into one that outlines research assumptions and objectives, while enhancing the background information and fundamental principles underlying the development and application of this strategy.

“Therefore, the objective of this study is to develop a Pt nanoparticle shell (PtNP-shell) with near-blackbody properties and ultra-high PCE in the NIR-II window (Fig. 1a) and investigate its potential in protecting against MI and myocardial reperfusion injury accompanying intervention through rapid, efficient, and precise multifunctional autonomic neuromodulation (Fig. 1b).”

Comment 3: *Similarly, the results section offers very little narrative or explanation of the results, which makes it difficult to follow the rationale of each experiment. For example, Vacht and c-fos staining are introduced with no explanation of what those are, why they are being stained, or what those results actually mean.*

Author reply: Thank you for your kind reminding. We have shown immunofluorescence staining in more details:

“Furthermore, immunofluorescence staining for vesicular acetylcholine transporter protein (VAChT), c-Fos, and TRPV1 on histopathological sections of photothermally modulated NGs served to localize parasympathetic neurons and reflect neuronal activity as well as TRPV1 protein expression, respectively (Fig. 4i).”

“Furthermore, immunofluorescence staining was conducted on LSG tissues to examine the expression of tyrosine hydroxylase (TH), c-Fos, and TREK1 (Fig. 6i). Localization of sympathetic neurons by TH proteins and reflection of neuronal activity as well as TREK1 protein expression.”

Comment 4: *The work appears to be original. The authors have previously used the same canine I/R models and stimulation of the NG and LSG by other means, but I do not find this NP formulation described previously. Statistical tests, sample sizes and P values are mostly clearly annotated and described, and seem appropriate for the types of data being analysed. Sample sizes also seem reasonable, though it not always clear which data points are replicates, independent biological samples, or actual experimental repeats.*

Author reply: To ensure the reproducibility and accuracy of our study, we repeated the experiment with independent biological samples *in vitro* and *in vivo* experiments to obtain the statistical information, including Fig 3d–j, Fig 4f–h, Fig 5e–k, Fig 6f–h, and Fig 7d–i, as well as the corresponding supplemental data.

Comment 5: *I do not find any major flaws with the NP synthesis or characterisation aspects, though this is not my specialty. However, I do have a few questions about the canine I/R model: The text mentions that “The NG was subsequently exposed to NIR-II laser irradiation for a duration of 5 minutes prior to occlusion of the left anterior descending (LAD) coronary artery for reperfusion therapy.” This text, and the diagram in Figures 4B, 5B and 6B, explain that the treatment was done before the I/R was induced. I am curious about why the authors choose a pre-treatment experimental design, rather than initiating the treatment after reperfusion. Obviously investigating only pre-treatment greatly lowers the clinical/therapeutic relevance of the research.*

Author reply: Thank you for the comment. The clinical management of acute myocardial infarction primarily involves transvascular interventions to restore blood flow in occluded vessels. However, reperfusion during this process can lead to acute ventricular arrhythmic events (*Mediat. Inflamm.* **2017**, 2017, 14). To mitigate these adverse effects through neuromodulation, we performed NIR-II irradiation intervention prior to reperfusion injury simulation, mimicking the clinical practice scenario. Furthermore, photothermal activation or inhibition of nerves exhibits sustained efficacy. This experimental protocol aligns more closely with the clinical significance of reducing reperfusion injury in patients undergoing myocardial infarction treatment.

Comment 6: *Related to this, why do the authors think there is a reduction of serum troponin and myoglobin? (It is also not clear at what time point these samples were taken. Does this reflect acute cardio protection?) In my opinion, if the authors want to say that the system is cardioprotective (line 292) or protects against cardiac damage (line 350), additional metrics (echocardiography, infarct volume etc) would be required to support this claim.*

Author reply: Thank you for the comment. Serum troponin and myoglobin are the primary and crucial clinical indicators of myocardial injury. Myoglobin levels typically increase within 1 to 2

hours following myocardial injury, while troponin elevation usually occurs 2 to 3 hours later. These two serum markers of cardiac injury can remain elevated for approximately 12 hours (*Int. J. Cardiol.* **2005**, *98*, 285). By implementing this experimental protocol, which involves collecting and testing blood samples at a time interval of 4 to 5 hours after modeling cardiac injury, accurate detection of myocardial injury in diverse individuals is facilitated.

Furthermore, significant differences in infarct area detection or ultrasound cardiac function testing can be observed in long-term myocardial ischemia models (*Adv. Sci.* **2023**, *10*, 2205551; *Basic Res. Cardiol.* **2022**, *117*, 34). Previous studies have usually indicated the extent of myocardial injury through acute indices, such as biomarkers of myocardial injury, in acute myocardial ischemia models (*Adv. Mater.* **2023**, *35*, 2304620). We utilized serum markers of myocardial injury to reflect the severity of acute myocardial infarction or acute reperfusion injury.

Comment 7: *Figure 4f and 4g are a little difficult to understand. The Y axis is different between the two graphs, but it seems that the effect on max HR is quite mild? I think it would make sense to put the baseline and laser irradiation groups on the same graph and statistically compare those too.*

Author reply: Thank you for the comment. We adjusted the Y axis of Fig. 4f (Fig. R2a) to align with Fig. 4g (Fig. R2b), facilitating a direct comparison of neural function between baseline and laser irradiation conditions. Considering our aim to demonstrate the neural functions of both the PtNP-shell group and control group under these two conditions, presenting the data separately would enhance clarity rather than combining them in one graph. This separate presentation allows for visualization that there is no significant difference between the two groups in the baseline condition, while highlighting that after laser irradiation, PtNP-shell significantly enhances neural activity across all levels of stimulation.

Fig. R2 | Maximal HR changes of beagle treatment with PtNP-shell or control a, before and b, after NIR-II exposure, n = 6.

Comment 8: *I am curious about the overall delivery efficiency, and biodistribution of the NPs and whether direct injection into the NG/LSG is clinically applicable.*

Author reply: The ganglion's surface is enveloped by a dense connective tissue matrix, facilitating our direct microinjection of PtNP-shell into the ganglion with exceptional efficiency and minimal loss. Long-term biosafety monitoring revealed an absence of NP distribution in histologic examinations of vital metabolic-immune organs such as the heart, liver, spleen, lungs, and kidneys, while liver and kidney functions remained unaffected (Fig. R3 and R4).

In clinical practice, ultrasound-guided injection of nerve blocking drugs into the ganglion enables local ganglion blockage (*Curr. Pain Headache Rep.* **2014**, *18*, 424). Our experimental approach aligns with established techniques for ganglion blocks and thus holds potential applicability in clinical settings.

Fig. R3 | Ganglion biocompatibility of targeted injections of PtNP or PBS after NIR-II irradiation and after 30 days of follow-up. a, Representative images of H&E and TUNEL staining of NG from different treatment groups immediately after NIR-II irradiation or after 30 days of follow-up. **b,** Representative images of H&E and TUNEL staining of LSG from different treatment groups immediately after NIR-II irradiation or after 30 days of follow-up.

Fig. R4 | Long term biosafety of PtNP-shell microinjection. Long-term in vivo biosafety was assessed by local injection of PtNP-shell into the ganglion of Beagle or by injection of equal doses of PtNP-shell into the tail vein of Sprague-Dawley rats. **a**, Representative H&E staining of major organs of beagles following different treatments. Blood biochemical analyses including **b**, ALT, **c**, AST, **d**, Urea, **e**, Crea, **f**, LDH1, **g**, TNF- α , and **h**, IL-6 were performed on Beagles in different treatment groups. **i**, Representative H&E staining of major organs of rats following different treatments. Blood biochemical analyses including **j**, ALT, **k**, AST, **l**, Urea, **m**, Crea, **n**, LDH1, **o**, TNF- α and **p**, IL-6 were performed on rats in different treatment groups.

Comment 9: *I also note that the in vivo comparisons are simply nanoparticles vs. PBS. This can clearly demonstrate that the NPs have some activity; but how do they compare to other methods of stimulating the NG/LSG, or drugs which stimulate the sympathetic/parasympathetic nervous system? Without these sorts of comparison, it is difficult to put the significance of authors' findings into context. If the authors can demonstrate that their approach is better than other approaches, this could be a lot more convincing.*

Author reply: Thank you for the comment. Currently, β -blockers are the primary pharmacological drugs employed in clinical practice for arrhythmia treatment (*J. Am. Heart Assoc.* **2018**, *7*, e007567; *Eur. Heart J.* **2023**, *44*, 3720). However, their administration during acute myocardial ischemia remains unclear and is contraindicated in patients with heart failure. Additionally,

our previous research investigated the local ganglion blockade using botulinum toxin A to protect the heart (*Heart Rhythm* **2022**, *19*, 2095). Nevertheless, its prolonged blocking effect renders it unsuitable for acute myocardial ischemia management. Conversely, PtNP-shell-based photothermal regulation offers reversible modulation within a short timeframe, exhibiting superior efficacy and controllability.

Cardiac sympathetic denervation (CSD) is a clinical procedure aimed at targeting the autonomic ganglia for refractory ventricular arrhythmias treatment. However, ganglion removal can be traumatic for patients and may lead to complications due to the loss of original physiological function (*Eur. Heart J.* **2022**, *43*, 2096).

Therefore, we aim to investigate the precise photothermal neuromodulation strategy facilitated by PtNP-shell as a reversible intervention approach for inhibiting ventricular arrhythmia associated with ganglion dysfunction. Notably, modulation specifically targeting NG has not been reported thus far, while non-invasive reversible neuromodulation strategies against LSG have primarily focused on photothermal modulation. Consequently, our study mainly established control and NP photothermal modulation groups similar to previous investigations (*Adv. Mater.* **2023**, *35*, 2304620; *Adv. Funct. Mater.* **2019**, *29*, 1902128).

Comment 10: *In terms of overall interest, I think is a good technical demonstration of clever system; but what is the real-world application? Could the authors envisage a way in which this technology can actually be applied to MI patients?*

Author reply: Thank you for the comment. Both myocardial ischemia and reperfusion injury-induced malignant arrhythmic events pose challenges in the treatment of acute myocardial infarction. Photothermal neuromodulation based on PtNP-shell offers a precise, transient, and reversible approach that may serve as an adjuvant therapy to improve patient prognosis. Additionally, ultrasound-guided targeted ganglionic microinjection is suitable for clinical applications. However, it is undeniable that despite the excellent tissue penetration depth of NIR-II photons (5–20 mm) (*Nat. Nanotech.* **2009**, *4*, 710; *Nat. Med.* **2012**, *18*, 1841; *Nat. Biomed. Eng.* **2017**, *1*, 0010), photothermal regulation still encounters challenges in intervening deeper tissues. Considering the presence of blood vessels surrounding the ganglion, which offers an opportunity to reduce the propagation path of photons in tissues, approaching the NIR-II fiber to the ganglion for photothermal modulation through a transvascular route during interventional therapy could be a potential solution for direct clinical translational application.

Comment 11: *The abstract mentions that the NPs conferred protection against ventricular arrhythmias following MI. However, supplementary figure 25 seems to show that there was no difference in overall VA events.*

Author reply: Thank you for the comment. The statistics of ventricular arrhythmia events encompass various forms, including ventricular premature beats (VPBs), ventricular tachycardia (VT), and ventricular fibrillation (VF). VT is defined as the occurrence of three or more consecutive VPBs. As depicted in Fig. R5 (Fig. 7i), the PtNP-shell group exhibited a significantly lower number of VPBs compared to the control group ($p < 0.05$).

Fig. R5 | Quantitative analysis of VF threshold in different groups. Data are shown as the mean \pm S.E.M. * $P < 0.05$, ** $P < 0.01$, *** $P < 0.001$.

Fig. R6 (Supplementary Fig. 28) presents a comparison between groups regarding the frequency and duration of VT occurrences. Although not statistically significant, there was a notable trend towards VT suppression in the NP group when compared to the control group. Supplementary Fig. 25 has been updated to Supplementary Fig. 28.

Fig. R6 | **Statistical analysis of recorded ventricular arrhythmia events post myocardial ischemia.** Quantitative analysis the number of **a**, VTs and **b**, the duration of sVT of beagles with MI. Data are shown as the mean \pm S.E.M. ns means that the difference is not statistically significant.

Comment 12: *Figure 3i is not very easy to read or understand.*

Author reply: Thank you for the comment. We conducted a repeated experiment and made modifications to fig3i, transforming the 3D diagram into two 2D diagrams for enhanced

comprehensibility. Fig. 3i (Fig. R7a) and Fig. 3j (Fig. R7b) depict cell viability following two different power density ($0.75 \text{ W}\cdot\text{cm}^{-1}$ and $1 \text{ W}\cdot\text{cm}^{-1}$) NIR-II laser irradiations at varying time intervals, respectively.

Fig. R7 | Effect of NIR-II laser irradiation with varying durations on the viability of HT-22 cells treated with PtNP-shell ($50 \mu\text{g}\cdot\text{mL}^{-1}$) (Power densities: **a**, $0.75 \text{ W}\cdot\text{cm}^{-2}$ and **b**, $1 \text{ W}\cdot\text{cm}^{-2}$). The error bar indicates S.E.M.

Comment 13: *Line 373, I think that more than a few blood tests and some organ histologically is required to make such a strong claim of “unequivocally demonstrate” long-term safety.*

Author reply: Thank you for the comment. In supplementary Figure 27, we first verified neuronal safety by histologic examination of the microinjected ganglion tissue as well as tunnel staining. In addition, we monitored the biosafety of both local tissue-injected beagles and tail vein-injected rats for 1 month. The major organs of these individuals were examined histologically, including the heart, liver, spleen, lungs, and kidneys. The most important routes of elimination, such as liver function and kidney function, were also examined by serology. In addition, we analyzed the effect of PtNP-shell on the inflammatory response in vivo by ELISA assay (Fig. R8). These detailed tests demonstrated the long-term biosafety of the NPs. The corresponding results have been included in the Supplementary Fig. 31g,h,o,p.

Fig. R8 | Blood biochemical analyses including **a**, TNF- α , and **b**, IL-6 were performed on Beagles in different treatment groups. Blood biochemical analyses including **c**, TNF- α and **d**, IL-6 were performed on rats in different treatment groups.

Comment 14: *Supplementary figure 6, 13. I think these would be more readable as tables rather than bar charts.*

Author reply: Thank you for the comment. We have changed Supplementary Fig. 6 and 13 into Supplementary Table 1 (Table R2) and Supplementary Table 2 (Table R3), respectively.

Table R2 | ξ potential of bare PtNP-shell, PtNP-shell + KOH and PtNP-shell@PEG.

	Zata Potential (mV)	St Dev(mV)
PtNP-shell	45.8	5.8
PtNP-shell + KOH	-25.7	4.64
PtNP-shell@PEG	-19.9	3.69

Table R3 | Comparison of photothermal conversion efficiency.

	Photothermal conversion efficiency (%)
This work	73.70
PEDOT:ICG@PEG-GTA ¹	71.10
MINDS ²	71.00
PTG NPs ³	67.60
RBC@Cu _{2-x} SeNPs ⁴	67.20
AuDAg ₂ S ⁵	67.10

MAPSULES ⁶	67.00
Fe ₃ O ₄ @PPy@GOD NCs ⁷	66.40
NPPBTPBF-BT ⁸	66.40
AS1064 ⁹	65.92
Gold Nanoraspberry ¹⁰	65.00
P-Pc-HSA ¹¹	64.70
Ultrathin polypyrrole nanosheets ¹²	64.60
H _x MoO ₃ ¹³	60.90
NiP PHNPs ¹⁴	56.80
FP NRs ¹⁵	56.60
COF ¹⁶	55.20
2MPT ⁺⁻ -CB ¹⁷	54.60
SPN-PT ¹⁸	53.00
Pdots ¹⁹	53.00
Pt Spirals ²⁰	52.50
TBDOPV-DT ²¹	50.50
Ti ₂ O ₃ @HA NPs ²²	50.20
TBDOPV-DT NP ²³	50.00
DPP-IID-FA NPs ²⁴	49.50
SPN-DT ¹⁸	49.00
NP ²⁵	49.00
FTQ nanoparticles ²⁶	49.00
CNPs ²⁷	49.00
H-SiO _x NPs ²⁸	48.60
BETA NPs ²⁹	47.60
CN-NPs ³⁰	47.60
Pt-NDS ³¹	46.90
MoO _{3-x} nanobelts ³²	46.90
P3 NPs ³³	46.00
Ni ₉ S ₈ ³⁴	46.00
PtAg nanosheets ³⁵	45.70
Nb ₂ C NSs ³⁶	45.65
V ₂ C-TAT@Ex-RGD ³⁷	45.10
PEG-TONW NRs ³⁸	43.60
SPNI-II ³⁹	43.40
1T-MoS ₂ ⁴⁰	43.30
Bi@C NPs ⁴¹	43.20
Au NPL@TiO ₂ ⁴²	42.10
CT NPs ⁴³	42.00
PPy-PEG NPs ⁴⁴	41.97
AuPt@CuS NSs ⁴⁵	41.56
Bi ₁₉ S ₂₇ I ₃ nanorods ⁴⁶	41.50
Cu ₃ Bi ₃ NR ⁴⁷	40.70
MPAE-NPS ⁴⁸	40.07

Comment 15: *Supplementary figure 7; this is quite a broad range of nanoparticle sizes. What makes up the smaller (100 nm) particles? Is there aggregation to produce larger particles?*

Author reply: Thank you for the comment. As show in Fig. R9, the smaller PtNP-shell (100 nm) is composed of ultra-small Pt nanoparticles (2–5 nm), similar to the larger PtNP-shell (200 nm).

Fig. R9 | Characterization of the smaller PtNP-shell (100 nm). a, TEM image and b, element mapping of the smaller PtNP-shell (100 nm).

PtNP-shell will not aggregate to produce larger particles. We surface-modified the PtNP-shell with PEG to enhance its biocompatibility and maintain its dispersibility in PBS. Additionally, dynamic light scattering analysis confirmed that the PtNP-shell exhibited little aggregation behavior after being undisturbed for 14 days (Fig. R10). The corresponding results have been included in the Supplementary Fig. 8.

Fig. R10 | Hydrodynamic size size of PtNP-shell after a, 1, b, 4, c, 7, and d, 14 days of standing (Inset: digital photograph).

To Referee #3

First of all, we really appreciate your constructive comments. According to your suggestions, we have made corresponding revisions to our manuscript as listed below.

Overall remark: *The manuscript describes photothermal neuromodulation via Pt nano-shell nanoparticles. Ga nanoparticles are used as a template for electrocoupling substitution-based synthesis of the Pt nano-shell. Using KOH wet etching, Ga core is etched and a Pt nano-shell structure is obtained. The rough surface topography of the Pt nano-shell structure allows the particles to exhibit high optical absorbance. It is claimed that these particles have one of the highest photothermal energy conversion efficiencies. The photothermal conversion of optical irradiation is then utilized for stimulation of target cells and tissues via temperature activated ion channels- TRPV1 and TREK1. The potential application of such photothermal modulation technique in regulating cardiac pulsing is demonstrated with regards to protecting against acute ventricular arrhythmias. However, the manuscript does not include proper controls to demonstrate that in-vivo photothermal modulation is achieved exclusively through the Pt nanoparticles. In addition, there are certain claims and results that need to be better corroborated to reach the scientific requirements of the journal. Therefore, I cannot recommend that this manuscript be accepted at Nature Communications in its current form.*

Author reply: Following your suggestion, we have added appropriate controls to demonstrate that *in-vivo* photothermal modulation is achieved exclusively through PtNP-shell. Furthermore, we have meticulously validated various propositions and findings with enhanced detail. These comments have been systematically addressed and resolved. For further information, please refer to the one-to-one reply corresponding to each comment.

Comment 1: *Abstract: It is claimed that “the autonomic nervous system plays a pivotal role in the pathophysiology of cardiovascular diseases.” This sentence is misleading since the dysregulation of the autonomic nervous system can contribute to cardiovascular diseases. However, it is not the primary contributor to the diseases, autonomous nervous system in fact regulates normal functioning of the cardiovascular system. (see: Purves D, Augustine GJ, Fitzpatrick D, LaMantia AS, McNamara JO, Williams SM. Autonomic regulation of cardiovascular function. Neuroscience. 2001;491-3 AND Gordan R, Gwathmey JK, Xie LH. Autonomic and endocrine control of cardiovascular function. World journal of cardiology. 2015 Apr 4;7(4):204.)*

Author reply: Thank you for the comment. We have made a modification: “Autonomic nervous system disorders play a pivotal role in the pathophysiology of cardiovascular diseases.”

Comment 2: *The authors claim that bi-directional reversible autonomic modulation is achieved via NIR-II photothermal modulation using Pt nano-shell nanoparticles. The manuscript presents uni-directional modulation where the target tissues are stimulated. Bi-directionality isn't demonstrated since in terms of neural interfaces, bi-directionality refers to the ability to record neural activity as well as stimulate (see: Song KI, Seo H, Seong D, Kim S, Yu KJ, Kim YC, Kim J, Kwon SJ, Han HS, Youn I, Lee H. Adaptive self-healing electronic epineurium for chronic bidirectional neural interfaces. Nature communications. 2020 Aug 21;11(1):4195. AND Hughes C, Herrera A, Gaunt R, Collinger J. Bidirectional brain-computer interfaces. Handbook of clinical neurology. 2020 Jan 1;168:163-81.).*

Author reply: Targeting the autonomic nervous system, we achieve nerve inhibition and activation through PtNP-shell-mediated photothermal effect, enabling adjustment of autonomic nervous imbalance. Hence, in this sense, we define it as a “bidirectional” neuromodulation.

Comment 3: *It is unclear why the NIR-II range was utilized in this work. This is important for selecting the right materials, models, and experiments. (see: nature.com/articles/s44222-023-00022-y AND nature.com/articles/s41551-022-00862-w).*

Author reply: Thank you for the comment. In comparison to the first near-infrared (NIR-I, 650–900 nm) and visible window, the photons in the second near-infrared window (NIR-II, 900–1700 nm) exhibits reduced tissue scattering and absorption, thereby increasing the maximum allowable exposure (MPE) of biological tissues. Consequently, photons within the NIR-II window exhibit significantly enhanced tissue penetration depths (up to 5–20 mm) (*Nat. Nanotech.* **2009**, *4*, 710; *Nat. Med.* **2012**, *18*, 1841; *Nat. Biomed. Eng.* **2017**, *1*, 0010). To achieve deep photothermal nerve regulation for cardioprotection, we developed PtNP-shell and validated its photothermal neuromodulation efficacy in the NIR-II window both *in vivo* and *in vitro*. Given its wavelength independence, further investigations may facilitate selection of a more suitable laser for achieving deeper tissue penetration while adhering to the MPE range. The exceptional potential of PtNP-shell makes it highly promising for precise neural regulation in deeper tissue and holds significant clinical translational value.

Comment 4: *Instead of using terminologies like “nearly perfect blackbody absorption”, the actual optical properties and metrics should be presented.*

Author reply: Thank you for the comment. Referring to the study (*Nat. Nanotech.* **2016**, *11*, 60), we have obtained the absorption spectra of PtNP-shell, which demonstrate its exceptional blackbody absorption characteristics. In the range of 250–1300 nm, the PtNP-shell exhibits an absorbance close to 1 at $75 \mu\text{g}\cdot\text{mL}^{-1}$, which is significantly enhanced in the range of 400–1300 nm compared to GaNPs and Ga@Pt NPs (Fig. R11). Additionally, optical images were acquired for comparison purposes.

Comparing with GaNPs and Ga@Pt NPs, the grayscale feature of PtNP-shell closely approximates the darkest point on the RGB spectrum, providing evidence that PtNP-shell exhibits a strong tendency towards perfect blackbody behavior (Fig. R12).

Fig. R11 | UV-vis-NIR absorption spectrum of GaNPs, Ga@Pt NPs and PtNP-shell ($75 \mu\text{g}\cdot\text{mL}^{-1}$).

Fig. R12 | Measurement of PtNP-shell blackness. **a**, Visual appearance of GaNPs, Ga@Pt NPs and PtNP-shell at different concentrations. **b**, Position of each color in the RGB cube, obtained by extracting the relative components of red, green and blue from Fig. R8a.

Comment 5: *Figure 1 presents how the nanoparticles will interact with the biological systems, however it does not show how light pulses/irradiation will be delivered to the target tissues/sites. This should be discussed in the figure and the manuscript since it is important for clinical translation.*

Author reply: Thank you for the comment. In Fig. 1 (Fig. R13), we have added the laser transmission path towards the target sites. Furthermore, the methodology section of the manuscript has a comprehensive account of the laser transmission process at the target sites.

“Initial vertical irradiation of NIR-II laser (1064 nm) at $0.80 \text{ W}\cdot\text{cm}^{-2}$ was performed on NG and LSG surfaces. The power density of the NIR-II laser was reduced to $0.45 \text{ W}\cdot\text{cm}^{-2}$ for continuous irradiation when the temperature of the NG reached $42.0 \text{ }^\circ\text{C}$, and was reduced to $0.6 \text{ W}\cdot\text{cm}^{-2}$ for continuous irradiation when the temperature of the LSG reached $46.0 \text{ }^\circ\text{C}$. The NIR-II laser irradiation remains stable with a spot size maintained at 1.0 cm^{-2} .”

Fig. R13 | The synthesis steps of the PtNP-shell and the concept of mediating precise photothermal effects for cardioprotection. a, The synthesis steps of PtNP-shell and schematic diagram of photothermal effect. **b,** Schematic diagram of multifunctional autonomic modulation mediated by photothermal effect of PtNP-shell for precise cardioprotection against myocardial I/R injury and MI-induced VAs.

Comment 6: Adequate controls should be provided to better compare the physical properties of PtNP-shells. That is, please provide the optical absorbance of GaNPs, Pt coated GaNPs for Figure 2.d; similar controls should be provided for Figure 2.e (including the thermal transients of such the solvent under irradiation).

Author reply: Following your suggestion, we supplemented the absorption spectra of GaNPs and Ga@Pt NPs in Fig. 2d (Fig. R14) and compared them with PtNP-shell. It was observed that the absorption of PtNP-shell in the range of 400–1300 nm was significantly higher than that of GaNPs and Ga@Pt NPs at equivalent concentrations. We made corresponding changes in the manuscript:

“In the range of 250–1300 nm, the PtNP-shell exhibits an absorbance close to 1 at $75 \mu\text{g}\cdot\text{mL}^{-1}$, which is significantly enhanced in the range of 400–1300 nm compared to GaNPs and Pt-coated Ga nanoparticles (Ga@Pt NPs) (Fig. 2d).”

Fig. R14 | UV-vis-NIR absorption spectrum of GaNPs, Ga@Pt NPs and PtNP-shell ($75 \mu\text{g}\cdot\text{mL}^{-1}$).

In addition, the thermal transient curves of PBS, GaNPs, Ga@Pt NPs and PtNP-shell are supplemented in Fig. 2e (Fig. R15). The results demonstrate that the rate at which PtNP-shell reaches the target temperature is significantly higher compared to that of GaNPs and Ga@Pt NPs. We made corresponding changes in the manuscript:

“Even in vitro, PtNP-shell ($50 \mu\text{g}\cdot\text{mL}^{-1}$) exhibited rapid temperature elevation, achieving a rise from room temperature to $41.0 \text{ }^\circ\text{C}$ and $45.0 \text{ }^\circ\text{C}$ within only 96 s and 133 s, respectively. However, for GaNPs (409 s and over 600 s) and GaIn@Pt NPs (317 s and over 600 s), it took significantly longer time to reach the same temperatures (Fig. 2e).”

Fig. R15 | Temperature elevation curves of PBS, GaNPs, Ga@Pt NPs and PtNP-shell ($50 \mu\text{g}\cdot\text{mL}^{-1}$) under NIR-II laser irradiation ($1 \text{ W}\cdot\text{cm}^2$).

Comment 7: *For the XPS characterization, a survey scan of presentative sample should be presented along with the detailed XPS characterization of oxygen (O1s) and potassium (K2p). The elemental composition of the PtNP-shells, Pt coated GaNPs, and GaNPs should be compared as well. This will better elucidate the composition of effectiveness of the synthesis protocols.*

Author reply: Following your suggestion, we supplemented the XPS survey spectrum of GaNPs, Ga@Pt NPs and PtNP-shell (Fig. R16), and detailed XPS characterization of oxygen (O1s) and potassium (K2p) (Fig. R17). The elemental compositions of GaNPs, Ga@Pt NPs and PtNP-shell were

compared by high-resolution XPS spectra (Fig. R17). We made corresponding changes in the manuscript and the results have been included in the Supplementary Fig. 5 and Supplementary Fig. 6.

Fig. R16 | The XPS survey spectrum of **a**, GaNPs, **b**, Ga@Pt NPs and **c**, PtNP-shell.

Fig. R17 | High-resolution XPS spectra and fitting results of **a**, GaNPs, **b**, Ga@Pt NPs and **c**, PtNP-shell.

“X-ray photoelectron spectroscopy (XPS) analysis reveals the presence of Ga and O in GaNPs, while Ga@Pt NPs and PtNP-shell exhibit the coexistence of Ga, O, and Pt (Supplementary Fig. 5). As depicted in the Supplementary Fig. 6, despite treatment with KOH, no presence of K element was detected in the PtNP-shell. The strong signals of Pt 4f_{5/2} and Pt 4f_{7/2} indicate that the Pt in Ga@Pt NPs and PtNP-shell exists in a metallic state (*Nat. Commun.* **2017**, *8*, 15802). In GaNPs, the peak centered at 1118.11 eV is attributed to Ga³⁺ 2p_{3/2}, while the peak centered at 1115.89 eV corresponds

to Ga 2p_{3/2}. In Ga@Pt NPs, the peak centered at 1118.80 eV is assigned to Ga³⁺ 2p_{3/2}, and the peak centered at 1116.52 eV corresponds to Ga 2p_{3/2}. As for PtNP-shell, the presence of a peak around 1118.56 eV indicates complete oxidation of Ga in PtNP-shell into Ga³⁺ (*Adv. Funct. Mater.* **2023**, *34*, 2302172). The O 1s spectrum was fitted using two peak functions, which were assigned to Ga–O at 530.44 eV (GaNPs), 530.98 eV (Ga@Pt NPs), 531.74 eV (PtNP-shell) and Ga–OH at 531.71 eV (GaNPs), 532.08 eV (Ga@Pt NPs), 532.74 eV (PtNP-shell) (*ACS Appl. Mater. Interfaces* **2024**, *16*, 4212). Compared to GaNPs and Ga@Pt NPs, the binding energy of the Ga–O and Ga–OH peaks in the PtNP-shell is shifted towards higher values, indicating a lower oxidation degree of the PtNP-shell.”

Comment 8: *The stability of the Pt-nanoshell suspensions should be evaluated as a function of time. Do the nanoparticle aggregate over time? Will this be a concern when the Pt-nanoshells are injected into biological systems.*

Author reply: Thank you for the comment. PtNP-shell exhibit long-term stability without aggregation, ensuring its compatibility for injection into biological systems. After standing for 1, 4, 7 and 14 days respectively, the statistically averaged hydrated nanoparticle size of PtNP-shell was determined using dynamic light scattering (Fig. R18). It is worth noting that the change of the average hydrated nanoparticle size of PtNP-shell remains negligible after 14 days, indicating its excellent stability. The corresponding results have been included in the Supplementary Fig. 8.

Fig. R18 | Hydrodynamic size of PtNP-shell after **a**, 1, **b**, 4, **c**, 7, and **d**, 14 days of standing (Inset: digital photograph).

Comment 9: *How does the addition of mPEG-SH5000 effect the photothermal properties of the nanoparticles?*

Author reply: Thank you for the comment. The current research indicates that the photothermal effect of mPEG-SH₅₀₀₀ itself can be disregarded (*ACS Nano* **2020**, *14*, 2265; *ACS Appl. Energ. Mater.* **2021**, *4*, 7710). We supplemented the temperature elevation curves of mPEG-SH₅₀₀₀ modified and unmodified PtNP-shell (Fig. R19a). The heating rate of the mPEG-SH₅₀₀₀ modified PtNP-shell is significantly higher compared to that of the unmodified PtNP-shell, potentially attributed to the agglomeration tendency of unmodified PtNP-shell at elevated temperatures, leading to a reduction in photothermal performance. Following 600 s of laser irradiation at 1064 nm, the statistically averaged hydrodynamic size for mPEG-SH₅₀₀₀-modified PtNP-shell was measured as 206.5 nm (Fig. R19b), whereas unmodified PtNP-shell exhibited a size of 1517 nm (Fig. R19c). TEM analysis further confirmed the observed agglomeration behavior in unmodified PtNP-shell subsequent to laser irradiation (Fig. R19d). The corresponding results have been included in the Supplementary Fig. 13.

Fig. R19 | The impact of PEG on the photothermal properties of PtNP-shell. a, Temperature elevation curves of SH-PEG modified and unmodified PtNP-shell. The hydrodynamic size of PtNP-shell b, before and c, after SH-PEG modification (after 600 s of 1064 nm laser irradiation). d, TEM image of PtNP-shell before SH-PEG modification (after 600 s of 1064 nm laser irradiation).

Comment 10: *Critical information from the methods section is missing- for example, details regarding the cell culture protocol and photothermal stimulation (such as power and pulse duration of optical irradiation are missing). How long was the ECG data recorded for? What were the exact stimulation conditions for all in-vivo experiments?*

Author reply: Thank you for your kind reminding. We provide a more detailed description of the Method:

“Cell-specific medium was prepared by mixing Dulbecco’s modified Eagle’s medium (DMEM), fetal bovine serum and penicillin-streptomycin mixture at 89%, 10% and 1%, respectively.”

“To induce activation of TRPV1 and TREK1 ion channels, which had been previously studied, (*Science* **2003**, *300*, 1284; *EMBO J.* **2000**, *19*, 2483) the culture dish was exposed to 1064 nm laser ($0.75 \text{ W}\cdot\text{cm}^{-2}$, TRPV1: 50 s, TREK1: 80 s), resulting in an elevation of temperature.”

“The same grouping pattern as in part1 was used, with NIR-II irradiation (Heating stage: $0.8 \text{ W}\cdot\text{cm}^{-2}$, 12 ± 3 s; Equilibrium stage: $0.45 \text{ W}\cdot\text{cm}^{-2}$, 5 min) of the NG before opening the occluded LAD coronary vessel.”

“The *in vivo* effects of precise photothermal stimulation of the sympathetic nervous system by PtNP-shell under NIR-II irradiation (Heating stage: $0.8 \text{ W}\cdot\text{cm}^{-2}$, 25 ± 5 s; Equilibrium stage: $0.6 \text{ W}\cdot\text{cm}^{-2}$, 5 min) were explored.”

“Malignant arrhythmic events occurring within 1 hour of MI and I/R injury were assessed by electrocardiographic recordings in a canine model using Lead 7000 Computerized Laboratory System.”

Comment 11: *For the in-vitro experiments, are the Pt nanoparticles engulfed by the target cells or are they localized in the vicinity of the cell membrane?*

Author reply: The PtNP-shell is partially localized within the target cells, while the remaining portion exhibits distribution around the cell membrane. We supplemented TEM and SEM images of PtNP-shell ($50 \mu\text{g}\cdot\text{ml}^{-1}$) co-cultured with cells for 24 hours. Cross-sectional TEM and SEM images showed that PtNP-shell was randomly distributed inside or on the surface of the cells (Fig. R20). This is attributed to the fact that the PtNP-shell exhibits a particle size of approximately 200 nm, enabling smaller particles to traverse the cell membrane. The corresponding results have been included in the Supplementary Fig.15.

Fig. R20 | PtNP-shell co-cultured with neurons. a and b, Cross-sectional TEM and c, SEM of the neurons incubated with PtNP-shell particles for 24 h.

Comment 12: *Both in-vivo and in-vitro photothermal stimulation experiments require the cells' microenvironment to reach temperatures greater than 42 °C. Does repeated photothermal stimulation using such high temperatures adversely affect cellular health by disrupting the cell membrane or trigger heat shock response?*

Author reply: The discovery of temperature-sensitive ion channels has given rise to a boom in the modulation of neuronal activity by temperature. Numerous experiments on temperature modulation of neuronal activity have shown that temperatures of 42° or even above can safely achieve reversible modulation of neural activity (*Nano Converg.* **2022**, 9, 13; *Brain Stimul.* **2021**, 14, 790). In addition, our results also indicate that the NPs-mediated photothermal modulation strategy is biologically safe, both at the cellular (Fig R21) and tissue levels (Fig R22 and R23).

Fig. R21 | **a**, Cell viability of HT-22 treated with different concentrations of PtNP-shell for 24 h. Effect of NIR-II laser irradiation with varying durations on the viability of HT-22 cells treated with PtNP-shell ($50 \mu\text{g}\cdot\text{mL}^{-1}$) (Power densities: **b**, $0.75 \text{ W}\cdot\text{cm}^{-2}$ and **c**, $1 \text{ W}\cdot\text{cm}^{-2}$). The error bar indicates S.E.M.

Fig. R22 | **Ganglion biocompatibility of targeted injections of PtNP or PBS after NIR-II irradiation and after 30 days of follow-up.** **a**, Representative images of H&E and TUNEL staining of NG from different treatment groups immediately after NIR-II irradiation or after 30 days of follow-up. **b**, Representative images of H&E and TUNEL staining of LSG from different treatment groups immediately after NIR-II irradiation or after 30 days of follow-up.

Fig. R23 | Long term biosafety of PtNP-shell microinjection. Long-term in vivo biosafety was assessed by local injection of PtNP-shell into the ganglion of Beagle or by injection of equal doses of PtNP-shell into the tail vein of Sprague-Dawley rats. **a**, Representative H&E staining of major organs of beagles following different treatments. Blood biochemical analyses including **b**, ALT, **c**, AST, **d**, Urea, **e**, Crea, **f**, LDH1, **g**, TNF- α , and **h**, IL-6 were performed on Beagles in different treatment groups. **i**, Representative H&E staining of major organs of rats following different treatments. Blood biochemical analyses including **j**, ALT, **k**, AST, **l**, Urea, **m**, Crea, **n**, LDH1, **o**, TNF- α and **p**, IL-6 were performed on rats in different treatment groups.

Comment 13: *The claim that Pt-NP shell does not induce significant damage to neurons under controlled NIR-II laser irradiation is incorrect since there is ~10% loss in cellular viability.*

Author reply: Thank you for your kind reminding. We conducted a repeated experiment. As illustrated in Fig. R24, although there was a slight decrease in cell activity across all laser irradiation groups compared to the control group (laser duration time of 0 s), no statistically significant differences were observed (all P > 0.05). Hence, we conclude that “Pt-NP shell does not induce significant damage to neurons under controlled NIR-II laser irradiation”.

Fig. R24 | Effect of NIR-II laser irradiation with varying durations on the viability of HT-22 cells treated with PtNP-shell ($50 \mu\text{g}\cdot\text{mL}^{-1}$) (Power densities: **i**, $0.75 \text{ W}\cdot\text{cm}^{-2}$ and **j**, $1 \text{ W}\cdot\text{cm}^{-2}$). The error bar indicates S.E.M.

Comment 14: *It will be recommended that the data presentation in Figure 3.i be changed since the details of the data are difficult to comprehend through a 3-D plot.*

Author reply: Thank you for your kind reminding. We conducted a repeated experiment and made modifications to Fig. 3i, transforming the 3D diagram into two 2D diagrams for enhanced comprehensibility. Fig. 3i (Fig. R25a) and Fig. 3j (Fig. R25b) depict cell viability following two different power density ($0.75 \text{ W}\cdot\text{cm}^{-1}$ and $1 \text{ W}\cdot\text{cm}^{-1}$) NIR-II laser irradiations at varying time intervals, respectively.

Fig. R25 | Effect of NIR-II laser irradiation with varying durations on the viability of HT-22 cells treated with PtNP-shell ($50 \mu\text{g}\cdot\text{mL}^{-1}$) (Power densities: **i**, $0.75 \text{ W}\cdot\text{cm}^{-2}$ and **j**, $1 \text{ W}\cdot\text{cm}^{-2}$). The error bar indicates S.E.M.

Comment 15: *For the in-vivo photothermal stimulation experiments, can similar affects be achieved without the presence of the Pt-nanoshell particles? Figure 4.d presents high temperature gradients for the surrounding tissue as well. Stimulation using infra-red radiation has been demonstrated previously, see: doi.org/10.1364/OL.30.000504 and doi.org/10.1117/1.2121772.*

Author reply: Thank you for the comment. First of all, the aim of our study was to achieve precise, rapid, and reversible neuromodulation for cardioprotection through NPs mediated conversion of light

energy into thermal energy. Non-invasive intervention was also the goal we pursued, so we chose to use NIR-II, which has a deeper penetration depth, as the light source. Due to the limitations of NIR itself and its inability to radiate energy to deep tissues, the effects of NIR on nerves are not clinically significant when viewed in isolation. The high-temperature gradient observed in Fig. 4d is only the temperature transfer to the fat, muscle, and other tissues around the ganglion, and does not affect the changes in nerve activity and function.

For experimental rigor, we also added changes in ganglion local temperature and neural function before and after NIR irradiation alone at the same parameters. We found limited local temperature elevation ($< 2\text{ }^{\circ}\text{C}$) and no significant changes in nerve function under NIR irradiation alone, including NG (Fig. R26) and LSG (Fig. R27). The corresponding results have been included in the Supplementary Fig.19 and 25.

Fig. R26 | Direct effect of NIR irradiation of NG. **a**, Local temperature curve of NG under NIR-II irradiation. **b**, Neural function of NG before and after NIR-II irradiation.

Fig. R27 | Direct effect of NIR irradiation of LSG. **a**, Local temperature curve of LSG under NIR-II irradiation. **b**, Neural function of LSG before and after NIR-II irradiation.

Comment 16: *The biosafety of Pt-nanoshell particles was evaluated after a rapid excision of the LSG and NG tissues. Can the authors comments on the long-term biosafety of the nanoparticles in passive (without photothermal stimulation) and active (with photothermal stimulation) states?*

Author reply: Thank you for the comment. In Fig. R28, we first verified neuronal safety by histologic examination of the microinjected ganglion tissue as well as TUNEL staining. We also showed by neurofunctional testing that PtNP photothermal stimulation is safe and reversible (Fig. R29 and R30). In addition, we monitored the biosafety of both local tissue-injected beagles for 1 month. These detailed tests demonstrated the long-term biosafety of the NPs (Fig. R31).

Fig. R28 | Ganglion biocompatibility of targeted injections of PtNP or PBS after NIR-II irradiation and after 30 days of follow-up. **a**, Representative images of H&E and TUNEL staining of NG from different treatment groups immediately after NIR-II irradiation or after 30 days of follow-up. **b**, Representative images of H&E and TUNEL staining of LSG from different treatment groups immediately after NIR-II irradiation or after 30 days of follow-up.

Fig. R29 | Effect of PtNP-shell photothermal stimulation of NG. Maximal HR changes of beagles treatment with PtNP-shell or control from 1 to 3 hours after NIR irradiation, n = 6. Data are shown as the mean \pm S.E.M. *P < 0.05, **P < 0.01, ns means that the difference is not statistically significant.

Fig. R30 | Effect of PtNP-shell photothermal inhibition of LSG. Maximal SBP changes of beagles treatment with PtNP-shell or control from 1 to 3 hours after NIR irradiation, n = 6. Data are shown as the mean \pm S.E.M. *P < 0.05, **P < 0.01, ns means that the difference is not statistically significant.

Fig. R31 | Long term biosafety of PtNP-shell microinjection. Long-term in vivo biosafety was assessed by local injection of PtNP-shell into the ganglion of Beagle or by injection of equal doses of PtNP-shell into the tail vein of Sprague-Dawley rats. **a**, Representative H&E staining of major organs of beagles following different treatments. Blood biochemical analyses including **b**, ALT, **c**, AST, **d**, Urea, **e**, Crea, **f**, LDH1, **g**, TNF- α , and **h**, IL-6 were performed on Beagles in different treatment groups. **i**, Representative H&E staining of major organs of rats following different treatments. Blood biochemical analyses including **j**, ALT, **k**, AST, **l**, Urea, **m**, Crea, **n**, LDH1, **o**, TNF- α and **p**, IL-6 were performed on rats in different treatment groups.

Comment 17: *Page 2, line 21: Please include examples and appropriate references for “conventional international procedures for MI.”*

Author reply: We have attached the reference after “Conventional international procedures for MI”, and the reference number is “3”.

Comment 18: *Page 4, line 16: Please change the word “encapsulated on” since Pt is not encapsulated on the surface of GaNPs. Pt is deposited onto of GaNP core then it encapsulates GaNP core.*

Author reply: Thank you for your kind reminding. We have changed the word “encapsulated on” in the manuscript to “deposited on”.

Finally, we want to thank the referees again for their constructive comments on our work, and we hope our newly revised manuscript can reach the quality expectation to be published in *Nature Communications*. Please find our revisions marked in red copy of the revised manuscript.

Reviewers' Comments:

Reviewer #1:

Remarks to the Author:

I have no additional comments.

Reviewer #2:

Remarks to the Author:

The authors have addressed many of the comments I gave, such as improving explanations, results narrative, adding supplementary tables and revising some figure layouts for clarity. I also appreciate the new NP characterisation data. However, I still think there are a few points which need to be addressed further:

Comment 5: Please clarify the exact timing of NP and NIR treatment in relation to the surgery. The text says "The NG was subsequently exposed to NIR-II laser irradiation for a duration of 5 minutes prior to occlusion of the left anterior descending (LAD) coronary artery for reperfusion therapy." This means the treatment is given before artery occlusion. However, the authors' response to my comment is talking about the importance of reducing reperfusion injury (which I agree is very important). However, if preventing reperfusion injury is the goal, why not induce the MI, then give the NIR treatment at the time of artery re-opening and reperfusion? This would simulate the clinical reality where an intervention could be given before, or during reperfusion. If the experimental design indeed is treating the animals before MI, this limitation needs to be clearly mentioned.

Comment 6: The time point of blood sampling for myoglobin and c-TnI measurement is still not clear in the manuscript. The text simply says "after MI and myocardial I/R injury" (line 736). Please specify the exact time points used for data in 5j and 5k.

I also disagree that reductions in these biomarkers is strong enough evidence to demonstrate cardioprotection, which is claimed multiple times throughout the text. This claim can only be made if there are functional tests or at least histological findings (i.e. reduced infarct size). I think claiming the reduction in acute VAs is fair enough, but "cardiac protection" strongly implies preservation of tissue and corresponding functional changes.

Lastly, the authors also did not provide any explanation for *how or why* these markers would be reduced by the NP/NIR treatment. The implication of lower circulating damage markers would be that there is less cardiac muscle loss - but there are no data to support this. Again, this is where functional metrics would be very useful. Still, as a principle, it's not exactly clear to me how the NP/NIR-II intervention would lower myoglobin/c-TnI release.

Comment 9-10: I think the responses to my comments are fine, but some of these points should go into the manuscript discussion section.

Reviewer #3:

Remarks to the Author:

The authors have addressed most of the reviewer comments. Adequate control experiments and results have also been provided.

However, there are a couple more concerns regarding the revised manuscript-

1. For neural interfaces, bidirectionality is defined as the ability and record and stimulate neural activity. Therefore, stimulating and inhibiting neural activity should not be defined as bidirectionality of neuromodulation. A more appropriate term will be multi-modal neuromodulation.

2. All peaks in the XPS spectra should be identified. For example, there is an emergence of peaks (at ca. 400 eV) other than Ga and O in the GaNP samples. The elemental composition should be assessed to better elucidate the chemical composition of each sample.

Reply to the referees

To Referee #1

Overall remark: *I have no additional comments.*

Author reply: We appreciate your positive evaluation on our manuscript.

To Referee #2

Overall remark: *The authors have addressed many of the comments I gave, such as improving explanations, results narrative, adding supplementary tables and revising some figure layouts for clarity. I also appreciate the new NP characterisation data. However, I still think there are a few points which need to be addressed further.*

Author reply: We appreciate your positive evaluation on our manuscript. We have made a point-by-point response to your comments and carefully revised the manuscript as you suggested. For your reference, please find our revisions marked in red color.

Comment 1: *Please clarify the exact timing of NP and NIR treatment in relation to the surgery. The text says “The NG was subsequently exposed to NIR-II laser irradiation for a duration of 5 minutes prior to occlusion of the left anterior descending (LAD) coronary artery for reperfusion therapy.” This means the treatment is given before artery occlusion. However, the authors’ response to my comment is talking about the importance of reducing reperfusion injury (which I agree is very important). However, if preventing reperfusion injury is the goal, why not induce the MI, then give the NIR treatment at the time of artery re-opening and reperfusion? This would simulate the clinical reality where an intervention could be given before, or during reperfusion. If the experimental design indeed is treating the animals before MI, this limitation needs to be clearly mentioned.*

Author reply: Thank you for the comment. Previously for the timing of the intervention in the reperfusion injury model we led to misunderstandings in the text and picture descriptions. In fact, the time point of neuromodulation by NIR irradiation was just 5 min before reperfusion injury, which is consistent with the timing of interventions for clinical application in ischemia-reperfusion injury.

We first performed microinjections of PtNP-shell in ganglia and occluded the left anterior descending (LAD) coronary artery for 1 h to induce myocardial ischemia. Subsequently, NIR-II laser irradiation was applied to NG for 5 min, followed by reperfusion treatment achieved by opening the ligated knot. Consequently, we further refined the corresponding content and Fig. 5b (Fig. R1) to provide a clearer and more direct representation of the treatment time.

Fig. R1 | Flowchart of regulating NG to protect against myocardial I/R injury and associated VAs.

“The left anterior descending (LAD) coronary artery was occluded for 1 h to induce myocardial ischemia. Subsequently, NIR-II laser irradiation was applied to NG for 5 min, followed by reperfusion treatment achieved by opening the ligated knot.”

Comment 2: The time point of blood sampling for myoglobin and c-TnI measurement is still not clear in the manuscript. The text simply says “after MI and myocardial I/R injury” (line 736). Please specify the exact time points used for data in 5j and 5k.

Author reply: Thank you for your kind reminding. Blood samples for myoglobin and c-TnI measurement were collected via jugular vein of each beagle 4-5 h after coronary artery ligation (3–4 h after reperfusion treatment). We have made corresponding changes in the manuscript.

“Serum Elisa assay revealed significantly lower levels of markers of myocardial injury (MYO and c-TnI) at 4–5 h post-infarction in the PtNP-shell group compared to the control group (all $p < 0.05$, Fig. 5j and k).”

In Methods:

“In myocardial I/R model experiments, 5 mL of venous blood was obtained from the jugular vein of each beagle 4–5 hours after ligation of the coronary vessels (3-4 h after reperfusion treatment).”

Comment 3: I also disagree that reductions in these biomarkers is strong enough evidence to demonstrate cardioprotection, which is claimed multiple times throughout the text. This claim can only be made if there are functional tests or at least histological findings (i.e. reduced infarct size). I think claiming the reduction in acute VAs is fair enough, but “cardiac protection” strongly implies preservation of tissue and corresponding functional changes.

Author reply: Thank you for the comment. We totally agree with your insightful suggestion, and have substituted the term “cardioprotection” with “reduction in the occurrence of ventricular arrhythmias induced by myocardial ischemia or reperfusion injury” in the manuscript.

In this study, we investigated the role of PtNP-shell-mediated photothermal neuromodulation in a myocardial infarction (MI) model and a myocardial ischemia/reperfusion (I/R) injury model. Our findings not only demonstrated a reduction in myocardial injury biomarkers but also revealed that the neuromodulation technique effectively improved cardiac electrophysiological stability, suppressed the occurrence of MI or I/R-induced VAs.

Indeed, serum markers of myocardial injury in acute infarction models and acute reperfusion injury models do not fully reflect cardioprotective effects. Comprehensive cardioprotective effects should be further assessed in the long-term myocardial injury model through its evaluation of cardiac function and infarct area indexes. The relevant content has been incorporated into the discussion

section of the manuscript.

“In this study, we validated the protective efficacy of PtNP-shell photothermal neuromodulation strategy in models of acute myocardial infarction and acute reperfusion injury to mitigate ventricular arrhythmia incidence. However, further evaluation through experiments such as assessment of cardiac function and infarct area is required to determine the cardioprotective potential of this strategy in chronic myocardial injury models.”

*Comment 4: Lastly, the authors also did not provide any explanation for *how or why* these markers would be reduced by the NP/NIR treatment. The implication of lower circulating damage markers would be that there is less cardiac muscle loss - but there are no data to support this. Again, this is where functional metrics would be very useful. Still, as a principle, it's not exactly clear to me how the NP/NIR-II intervention would lower myoglobin/c-TnI release.*

Author reply: Thank you for the comment. In the case of myocardial cell injury, biomarkers such as troponin, indicative of myocardial damage, are released into the bloodstream (*JAMA*. **2013**, 309, 2262). In the guidelines for cardiovascular disease published by the European Society of Cardiology and others, testing for cardiac injury biomarkers is also an important indicator for clinical detection of myocardial injury in patients (*Eur. Heart J.* **2012**, 33, 2551; *Eur. Heart J.* **2023**, 44, 3720). Therefore, we validate the protective efficacy of PtNP-shell mediated photothermal neuromodulation strategy against ischemia and reperfusion injury by assessing the levels of serum myocardial injury markers.

Neurotransmitters released by sympathetic nerves can bind adrenergic receptors in cardiomyocytes to control cardiomyocyte contraction (*Annu. Rev. Physiol.* **2022**, 84, 285). Myocardial ischemia causes activation of sympathetic nerves, releasing large amounts of sympathetic neurotransmitters (*Eur. Heart J.* **2024**, 45, 669). Subsequently, sympathetic activation of adrenergic receptors promotes sarcoplasmic reticulum calcium release from cardiomyocytes, exacerbating calcium overload and causing cardiac electrophysiologic disturbances (*J. Am. Coll. Cardiol.* **2010**, 56, 805). Acute adrenergic receptor activation results in rapid activation of cardiomyocyte-specific inflammatory vesicles, which induces IL-18 activation, promotes cardiac cytokine waterfall response and macrophage infiltration, and leads to myocardial injury and reduced cardiac function (Fig R2) (*Eur. Heart J.* **2018**, 39, 60). Additionally, parasympathetic nerve stimulation can elicit the activation of α -7 nicotinic acetylcholine receptor (α 7nAChR), leading to a reduction in inflammatory response and oxidative stress (*J. Am. Heart Assoc.* **2023**, 12, e030539). The activation of α 7nAChR has been shown to reverse the up-regulation of myocardial arginase induced by ischemia-reperfusion injury and reduce infarct size (Fig R3) (*Acta Physiol.* **2017**, 221, 174).

Fig. R2 | Take home figure working model for b-adrenergic activation induced cardiac inflammatory cascade which finally results in cardiac remodeling (left) and therapeutic strategy (right) (*Eur. Heart J.* **2018**, 39, 60).

Fig. R3 | Myocardial area at risk and infarct size. **a**, Area at risk expressed as % of left ventricle and **b**, infarct size (with representative images of infarcted myocardium) expressed as % of the area at risk following 30 min ischaemia and 2-h reperfusion in control animals (Control IR; n = 14), after vagal nerve stimulation (VNS + IR; n = 13), α_7 nAChR blockade and VNS (MLA + VNS + IR; n = 7), the arginase inhibitor nor-NOHA and IR (nor-NOHA + IR, n = 5), nor-NOHA+VNS+IR (n = 6) and MLA alone (n = 5). Data are shown as mean \pm SEM. Significant differences between groups are shown; *P < 0.05 and ‡P < 0.001 (*Acta Physiol.* **2017**, 221, 174).

In summary, neuromodulation (inhibition of sympathetic nerve or activation of parasympathetic nerve) may reduce myocardial injury and decrease serum levels of markers of myocardial injury through mechanisms such as reduction of calcium overload, inflammatory response, and oxidative stress. The corresponding content have been included in the manuscript.

“Serum Elisa assay revealed significantly lower levels of markers of myocardial injury (MYO and c-TnI) at 4–5 h post-infarction in the PtNP-shell group compared to the control group (all $p < 0.05$, Fig. 5j and k), indicating an improvement in myocardial injury (*JAMA*. **2013**, *309*, 2262). This may be attributed to the activation of α -7 nicotinic acetylcholine receptor by stimulating parasympathetic nerves, thereby alleviating inflammatory reactions and oxidative stress (*J. Am. Heart Assoc.* **2023**, *12*, e030539; *Acta Physiol.* **2017**, *221*, 174).”

Comment 5: Comment 9-10: I think the responses to my comments are fine, but some of these points should go into the manuscript discussion section.

Author reply: Thank you for your kind reminding. Your valuable suggestion has significantly enhanced the depth of our research. The relevant content has been incorporated into the discussion section of the manuscript.

“Cardiac sympathetic denervation (CSD) is a clinical procedure aimed at targeting the autonomic ganglia for refractory ventricular arrhythmias. However, ganglion removal can be traumatic for patients and may lead to complications due to the loss of original physiological function (*Eur. Heart J.* **2022**, *43*, 2096). Currently, β -blockers are the primary pharmacological drugs employed in clinical practice for arrhythmia treatment (*J. Am. Heart Assoc.* **2018**, *7*, e007567; *Eur. Heart J.* **2023**, *44*, 3720). However, their administration during acute myocardial ischemia remains unclear and is contraindicated in patients with heart failure. Additionally, previous research investigated the local ganglion blockade using botulinum toxin A to protect the heart (*Heart Rhythm* **2022**, *19*, 2095). Nevertheless, its prolonged blocking effect renders it unsuitable for acute myocardial ischemia management. Conversely, PtNP-shell-based photothermal neuromodulation offers reversible modulation within a short timeframe, exhibiting superior efficacy and controllability.”

“Simultaneously, exploiting the presence of blood vessels surrounding the ganglion presents an opportunity to minimize photon propagation within tissues. Consequently, photothermal modulation of NIR-II fibers in proximity to the ganglion through vascular routes during interventional therapy emerges as a promising avenue for direct clinical translation.”

To Referee #3

Overall remark: *The authors have addressed most of the reviewer comments. Adequate control experiments and results have also been provided. However, there are a couple more concerns regarding the revised manuscript.*

Author reply: We appreciate your positive evaluation on our manuscript. We have made a point-by-point response to your comments and carefully revised the manuscript as you suggested. For your reference, please find our revisions marked in red color.

Comment 1: *For neural interfaces, bidirectionality is defined as the ability and record and stimulate neural activity. Therefore, stimulating and inhibiting neural activity should not be defined as bidirectionality of neuromodulation. A more appropriate term will be multi-modal neuromodulation.*

Author reply: Thank you for the comment. We have substituted the term “bi-directional” with “multimodal”.

Comment 2: *All peaks in the XPS spectra should be identified. For example, there is an emergence of peaks (at ca. 400 eV) other than Ga and O in the GaNP samples. The elemental composition should be assessed to better elucidate the chemical composition of each sample.*

Author reply: Following your suggestion, we identified all peaks in the XPS survey spectra (Fig. R4). The peaks at about 400 eV can be identified to Auger peaks of Ga. The corresponding results have been included in the Supplementary Fig. 5.

Fig. R4 | The XPS survey spectra of **a**, GaNPs, **b**, Ga@Pt NPs and **c**, PtNP-shell.

Finally, we want to thank the referees again for their constructive comments on our work, and we hope our newly revised manuscript can reach the quality expectation to be published in Nature Communications. Please find our revisions marked in red copy of the revised manuscript.

Reviewers' Comments:

Reviewer #2:

Remarks to the Author:

The authors have addressed all comments in the latest version of the manuscript.

Reviewer #3:

Remarks to the Author:

The authors have addressed all comments.